# A simplified approach to failure analysis of ball bearings combining Principal Component Analysis and Fast Fourier Transform

Emmanuel Ndongue Esseme[1], Thomas Florent Kanaa [1,2,4]*, Thérèse Jacquie Ngo Bissé[2,3‡], Mathieu Jean Pierre Pesdjock[3‡], Ludovic Ngongang[1,2‡], Emmanuel Tonye[4‡]

**1** Laboratory of Mechanics, Department of Mechanical Engineering, Higher Technical Teacher's Training College, University of Douala, Douala, Cameroon, **2** Laboratory of Mechanics and Materials, Department of Mechanical Engineering, Higher Technical Teacher's Training College, University of Ebolowa, Ebolowa, Cameroon, **3** Cameroonian Association for Research and Innovation in Energy, Technology and Environment, Ebolowa, Ebolowa, Cameroon, **4** Laboratory of Electrical Engineering, Mechatronics and Signal Processing, Department of Electrical and Telecommunications Engineering, National Advanced School of Engineering of Yaounde, University of Yaoundé, Yaoundé, Cameroon

☯ These authors contributed equally to this work.
‡ TJNB, MJPP, LN, and ET are also contributed equally to this work.
* t_kanaa@yahoo.fr

## Abstract

Ball bearing monitoring employs time-frequency techniques to facilitate the early detection of faults; however, the presence of non-stationary or noisy signals can limit the effectiveness of these techniques, requiring advanced methods for reliable predictive maintenance. This study proposes a methodology for fault detection in complex systems, utilising Principal Component Analysis (PCA) to identify indicators with a higher probability of fault. Subsequent to this, the signal characteristics are decomposed using the Fast Fourier Transform (FFT). This technique is employed to identify the Hotelling component and the SPE (quadratic prediction error), with the objective of determining the state of health of the rolling bearings. This is achieved by extracting the frequencies and harmonics that characterise the fault. The Hotelling component considers elements in the main space with a higher energy representation for evaluation, while the SPE considers elements in the residual space. The results demonstrate a rapidly appreciable range of detection and dispersion of faulty signals. A comparative analysis of the KPCA-FFT and PCA-FFT results is performed. However, this study demonstrates that the combination of PCA-FFT with the Hotelling index test and SPE is more suitable for evaluating signals with defects.

## 1 Introduction

Bearings are extensively used in a while of mechanical systems, including those used in automotive, aerospace, machine tool and energy (i.e wind power, power

**Data availability statement:** The data employed in the manuscript for the analysis of rolling defects is sourced from the database, accessible via the following Uniform Resource Locator (URL) (reference [57]): https://engineering.case.edu/bearingdatacenter/12k-drive-end-bearing-fault-data.

**Funding:** The author(s) received no specific funding for this work.

**Competing interests:** The authors have declared that no competing interests exist.

transmission,hydraulics etc.). Therefore, They must perform precise reliability. However, given the extreme working conditions to which they are subjected, an unexpected early failure of these bearings can result in mechanisms breakdown, which can lead to damages, considerable economic losses and serious accidents [1]. It is of greater significance to note that less than 30% of bearings reach their fatigue limit and subsequently fail [2]. The elevated expense of wind energy systems can be attributed to premature bearing failure, which curtails the operational lifespan of gearboxes from 20 years to a mere 2 or 11 years. This, in turn, results in escalated maintenance expenditures amounting to up to 300,000 euros [3,4]. In the petrochemical industry, these failures are directly linked to 52% of engine failures on platforms [5,6]. Hence, it is of the utmost importance to monitor the performance of mechanical systems in general and the state of health of bearings in particular. [7–9] categorise bearing anomalies as either localised or distributed faults. It is observed that localised defects are responsible for 90% of bearing failures,and so it is crucial to control the dominant failure mode, contact fatigue, which is spalling [10–13]. In complex and challenging working environments, malfunctions frequently occur, resulting in breakdowns that are often obscured by background noise, particularly in the initial stages of damage. Deteriorating operating conditions have also been demonstrated to cause failures in related components. These failures can result in coupled bearing failures, which in turn lead to economic losses. In the context of this study, it is imperative to perform a thorough diagnosis of bearing defects, taking into account interference factors. Furthermore, it is imperative to diagnose bearing defects by taking into account interference factors and conducting research on the analysis of failure mechanisms. This is of crucial practical importance and operational technical value [14]. Consequently, the monitoring of ball bearings is predicated on time and frequency techniques that are designed to detect any faults at an early stage. Nevertheless, the inherent complexity of the signals (non-stationary, noisy) serves to limit the efficacy of these conventional methodologies. Consequently, there is a necessity for the implementation of more advanced approaches, such as three-dimensional analysis (time-frequency-amplitude), in order to characterise transient phenomena and modulations. The integration of intelligent algorithms, including advanced signal processing and learning models, has been demonstrated to enhance the interpretation of vibration signatures and improve the reliability of diagnosis. The objective of these innovations is to optimise predictive maintenance by integrating multidimensional representation spaces. The diagnosis of faults by machine learning has been the subject of a number of research projects. These have included the use of support vector machines and neural networks for the purpose of compressing the signal and reducing its dimensionality [15–20]. Another method frequently used in the reduction of data dimensions, such as the principal component analysis (PCA) method, was used by [21], for the identification of failures in photovoltaic panels through the integration of failure indicators developed using the Kullback-Leibler divergence [22]. In contrast, [23] employ Principal Component Analysis (PCA) and the Gaussian mixed model for the detection of cracks on centrifugal pump blades. [24] utilise a database obtained for this purpose to deploy SVM, the neural network method with radial basis function, PCA and Kernel Principal Component Analysis (KPCA) for the

detection of air handling unit failure and the prevention of reduced air quality in operating rooms.In the food industry, the detection of leaf tomato diseases is facilitated by the use of deep neural networks in conjunction with PCA, which has been demonstrated to yield excellent classification accuracy [25,26]. There are several methods for detecting bearing defects in vibration analysis based on the time, frequency and time-frequency domains. These have been proposed by various researchers [27–29]. In the time-scale domain, [30–32] demonstrate the effectiveness of the Hilbert-Huang transform in detecting low-amplitude signals, while the continuous wavelet transform is more suited to identifying the location of defects.[33,34] propose a hybrid method based on the Morlet wavelet filter to eliminate spurious signals. The output signal is then processed by an improved autocorrelation algorithm.[35] present the discrete wavelet transform as an ideal tool for analysing signals of transient or non-stationary nature. They find that the pulses appear periodically with a period of time corresponding to the characteristic frequencies of the faults. [36,37] propose Wavelet Multiresolution Analysis (WMRA) as an indicator sensitive to bearing defects in the presence of intense noise.[38] employed a combination of wavelet packet decomposition for feature extraction and Chaotic Sparrow Search Optimization Algorithms (CSSOA) to optimize the parameters of a Deep Belief Network (DBN). [39] utilized a multivariate algorithm of detection in space, the reliability of which was confirmed by an univariate analysis of variance on selected characteristics in the time domain; They then performed PCA and used a Mahalanobis index.[40] employ PCA and random matrix theory (RMT) to offset the loss of valuable information resulting from conventional feature extraction techniques. In recent years, a series of studies have been devoted to exploring the use of these methods for diagnosing bearing defects. In their study, [41] integrate weighted PCA with the Gaussian mixing model to enhance the diagnostic accuracy for defective bearings. [42], propose a rolling fault diagnostic model combining hybrid deep neural network, with principal component analysis to ensure the robustness of the algorithm, in the face of extremely variable loads. [43,44], presents a fault detection and diagnostic approach based on the combination of PCA and Kohonen algorithms on data residues.[45] decomposes the signal into multiple layers in order to identify incipient defects and extract background noise, employing Deep EMD-PCA $T^2$ and Squared Prediction Error (SPE) statistical index tests.[46] employed Hotelling's projected space to develop three bearing degradation monitoring indicators, designated $SDHT^2$, $VSDHT^2$, and $NVSDHT^2$. [47,48] combined the PCA with $T^2$ to determine the remaining life of the bearings. In order to solve the problem of the low accuracy of remaining useful life (RUL) prediction for motor rolling bearings, [49] propose a neural network model based on the Weibull proportional hazards model (WPHM) and stochastic configuration networks (SCNs), combined with the KPCA. [50] utilise KPCA to construct a nonlinear map of the raw feature space of vibration signals to a high-dimensional feature space. Subsequently, the statistical indices $T^2$ and SPE are employed to extract the feature vector of the fault signal. When the k-nearest neighbour method is employed in combinations with this approach, the classification accuracy rate is elevated to 96.67%. Given the proven effectiveness of these methods in the analysis of vibrations, they have often been used in conjunction with a single statistical indicator, resulting in the under utilisation of other potentially relevant indices. This limitation can result in an incomplete characterisation of components resulting from nonlinear transformations, particularly those associated with kernel functions. Despite the considerable progress that has been made in this field, only a small number of approaches have been developed that combine Kernel Principal Component Analysis (KPCA) and several statistical index tests for the diagnosis of bearing faults. Furthermore, the joint exploitation of the two subspaces generated by PCA or KPCA, namely the principal space and the residual space, remains relatively unexplored, although it could potentially significantly increase the ability to discriminate between defective and non-defective vibration signals. This paper presents an innovative and efficient method for the detection of bearing defects by vibration analysis. The primary innovation of this study lies in the development of a hybrid approach to diagnosing bearing faults. This approach combines dimension reduction techniques (ACP and KPCA), frequency analysis (FFT), and statistical evaluation (Hotelling $T^2$ tests and SPE index). The proposed methodology is based on the combined use of Principal Component Analysis (PCA) and its nonlinear extension, known as Kernel Principal Component Analysis (KPCA), in the analysis of vibration signals. This step allows for structural filtering of the data by decomposing it into two orthogonal subspaces:

- The principal space is characterised by the presence of high-variance components, which represent the dominant information in the signal.
- The residual space is characterised by its low-variance components, which frequently exhibit discrete anomalies or are obscured by noise. However, these components often possess significant potential.

In each of these subspaces, a Fast Fourier Transform (FFT) is applied to enhance the discriminating power and highlight the spectral components characteristic of defects. Subsequently, Hotelling's $T^2$ statistical indices (in the main space) and SPE (in the residual space) are evaluated, thus facilitating a dual diagnostic reading of the vibration behaviour. In summary, this innovative approach offers several notable advantages for the industrial sector:

- The utilisation of both information spaces enables a more comprehensive evaluation of the bearing condition.
- The efficacy of the approach is demonstrated by its ability to enhance the detection of defects of a subtle nature, such as those exhibiting low amplitudes or high noise levels, which frequently elude detection by conventional methodologies.
- The system has been developed to enhance monitoring sensitivity by adapting to the constraints of real industrial environments.

This section presents recent and relevant work in the literature and the methodology that will be used. The rest of the paper is structured as follows: Sect 2 presents the mathematical methodology and the description of the implementation. Sect 3 presents an analysis of the results and Sect 4 concludes the paper.

## 2 Methods

### 2.1 Description

Fig 1 presents a descriptive account of the proposed methodology. This figure offers a straightforward overview of the diagnostic process for bearing spalling defects, utilising two statistical indicators. Two analytical spaces are employed, one with residual components and the other with components of significant elements. The two components are derived through PCA, and the generated data is returned in frequency space using an FFT.

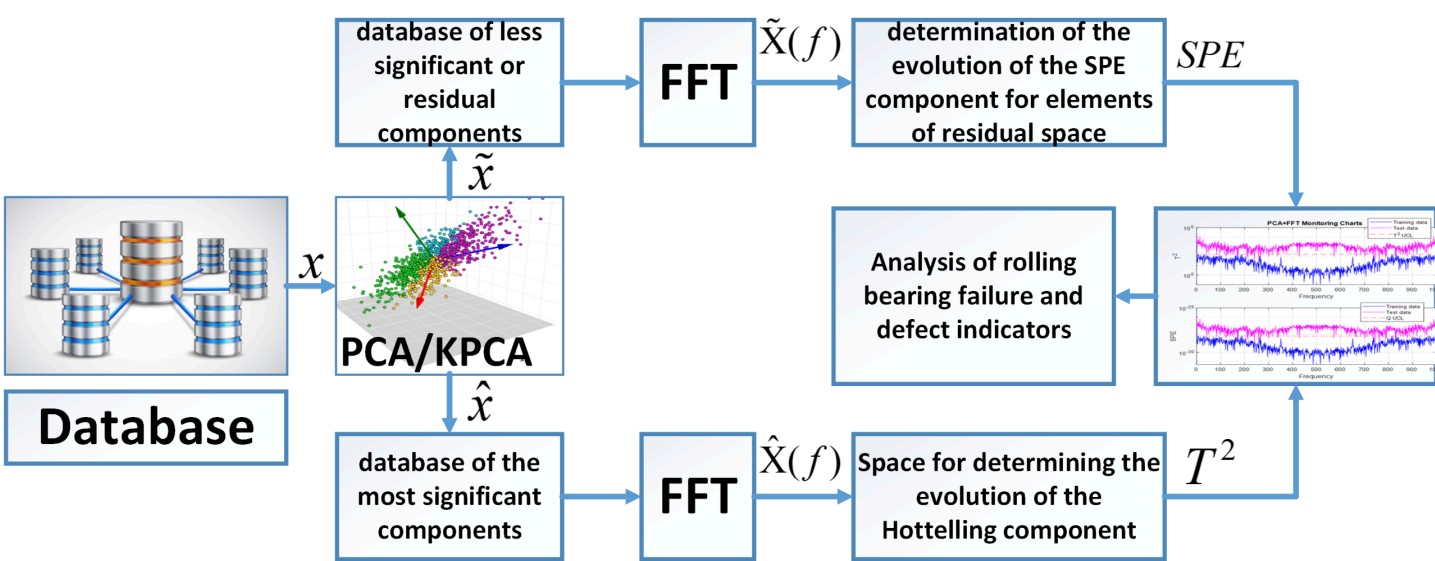

**Fig 1**. Description of the suggested fault vibration analysis protocol for rolling bearings.

## 2.2 Principal component analysis space design

Let $x(n,m)$ be the matrix extracted from the vibration database, as defined in Eq (1), where $n$ and $m$ correspond, respectively, to the number of observations and variables. Within the theoretical framework of the PCA method, this matrix forms the basis for the subsequent stages of normalisation and dimensionality reduction.

$$x = \begin{pmatrix} x_{11} & \cdots & x_{1m} \\ \vdots & \ddots & \vdots \\ x_{n1} & \cdots & x_{nm} \end{pmatrix} \tag{1}$$

In order to achieve data normalisation, the matrix defined in Eq (2) is rewritten and designated the reduced centred matrix of Eq (1).

$$X = \begin{pmatrix} \dfrac{x_{11} - \overline{x}_1}{\sigma_1^2} & \cdots & \dfrac{x_{1m} - \overline{x}_m}{\sigma_m^2} \\ \vdots & \ddots & \vdots \\ \dfrac{x_{n1} - \overline{x}_1}{\sigma_1^2} & \cdots & \dfrac{x_{nm} - \overline{x}_m}{\sigma_m^2} \end{pmatrix} \tag{2}$$

In accordance with the aforementioned equations, the mean and variance values are represented by $\overline{x}_j$ and $\sigma_j^2$, respectively.

$$\overline{x}_j = \frac{1}{n} \sum_{i=1}^{n} x_{ij}, \quad j = 1, 2, \cdots, m \tag{3}$$

$$\sigma_j^2 = \frac{1}{n} \sum_{i=1}^{n} \left( x_{ij} - \overline{x}_j \right)^2, \quad j = 1, 2, \cdots, m \tag{4}$$

To estimate the parameters of the PCA model, we must diagonalise the correlation matrix. This allows us to determine the matrices linking the eigenvalues and eigenvectors. An optimal linear transformation is given as follows [51].

$$t = P^T X \tag{5}$$

$t \in \mathbb{R}^m$ is the vector of principal components, $P = [p_1, p_2, \cdots, p_m] \in \mathbb{R}^{m \times m}$ is the matrix of eigenvectors $P_j; j = 1, 2, \cdots, m$ corresponding to the eigenvalues $\lambda_j$ obtained from the decomposition of the correlation matrix of $X$. The advantage of the relationship (5) is that it reduces the dimensionality of the presentation space by the number $l$ of principal components to be retained, resulting in the following partitioning.

$$P = [\hat{P} | \tilde{P}], \quad \hat{P} \in \mathbb{R}^{m \times l} \quad and \quad \tilde{P} \in \mathbb{R}^{m \times (m-l)} \tag{6}$$

In accordance with the specifications outlined in Eq (6), the vector $t$ is divided into two distinct components: the estimated vector $\hat{t}$ and error vector $\tilde{t}$.

$$t = \hat{t} + \tilde{t} \tag{7}$$

Where

$$\begin{cases} \hat{t} & = t\hat{\Pi}, \quad \hat{\Pi} = \hat{P}\hat{P}^T, \\ \tilde{t} & = t(Y - \hat{\Pi}) = t\tilde{\Pi}, \\ Y & = PP^T = P^T P. \end{cases} \tag{8}$$

Given the symmetry between the $\hat{\Pi}$ and $\tilde{\Pi}$ vectors, the vector of estimates must satisfy the following Equation [52].

$$\hat{\mathbf{t}}\tilde{\mathbf{t}}^T = \mathbf{t}^T \hat{\Pi}(Y - \hat{\Pi}\mathbf{t}) = 0 \tag{9}$$

The model can be decomposed into two orthogonal subspaces, as illustrated in Fig 2.

## 2.3 FFT space design

In the context of enhancing the discriminatory capability of principal component analysis (PCA) in the domain of vibration signals, a Fast Fourier Transform (FFT) is employed as a key component of the optimisation process. Frequency analysis of the signals enables the distinctive signatures of their spectrum to be highlighted. It is widely accepted that these signatures are more effective in determining fault conditions. Consider $t_{zj} = \hat{t}_{zj} + \tilde{t}_{zj}$, the elements representing the column vectors $j = 1, 2, \cdots, m$ from 7. Applying the fast Fourier transform [53] leads to:

$$T_{kj}\Big|_{k=1,\cdots,n} = \mathcal{F}\{\hat{t}_{zj}\} + \mathcal{F}\{\tilde{t}_{zj}\} = \sum_{z=1}^{n} t_{zj} w_n^{-zk} = \underbrace{\sum_{z=1}^{n} \hat{t}_{zj} w_n^{-zk}}_{\text{principal component}} + \underbrace{\sum_{z=1}^{n} \tilde{t}_{zj} w_n^{-zk}}_{\text{residual compoment}}$$

$$where \quad w_n = e^{\frac{2\pi}{n}\xi}, \quad and \quad \xi^2 = -1 \tag{10}$$

It is possible to divide the sum in such a way that the odd and even terms are grouped together, under the assumption that $n$ is a power of 2.

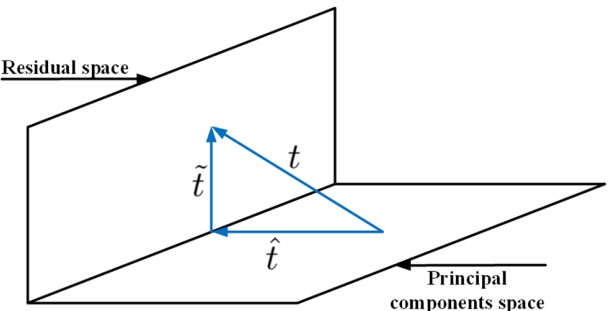

**Fig 2. Vector $t$ decomposes.**

$$T_{kj} = \sum_{z=1}^{\frac{n}{2}} \hat{t}_{2zj} w_n^{-2zk} + \sum_{z=1}^{\frac{n}{2}} \hat{t}_{(2z+2)k} w_n^{-(2z+2)k} + \underbrace{\sum_{z=1}^{\frac{n}{2}} \tilde{t}_{2zj} w_n^{-2zk} + \sum_{z=1}^{\frac{n}{2}} \tilde{t}_{(2z+2)k} w_n^{-(2z+2)k}}_{\text{residual component}}$$

$$= \underbrace{\sum_{z=1}^{\frac{n}{2}} \hat{t}_{2zj} w_n^{-2zk} + w_n^k \sum_{z=1}^{\frac{n}{2}} \hat{t}_{(2z+1)k} w_n^{-(2z+1)k}}_{\text{principal component}} + \underbrace{\sum_{z=1}^{\frac{n}{2}} \tilde{t}_{2zj} w_n^{-2zk} + w_n^k \sum_{z=1}^{\frac{n}{2}} \tilde{t}_{(2z+1)k} w_n^{-(2z+1)k}}_{\text{residual component}}$$

$$= \underbrace{\underbrace{\left(\mathfrak{I}_{\frac{n}{2}}(\hat{t}_{zj})\right)_{2k}}_{\text{even}} + \underbrace{w_n^k \left(\mathfrak{I}_{\frac{n}{2}}(\hat{t}_{zj})\right)_{(2z+1)}}_{\text{odd}}}_{\text{principal component}} + \underbrace{\underbrace{\left(\mathfrak{I}_{\frac{n}{2}}(\tilde{t}_{zj})\right)_{2k}}_{\text{even}} + \underbrace{w_n^k \left(\mathfrak{I}_{\frac{n}{2}}(\tilde{t}_{zj})\right)_{(2z+1)}}_{\text{odd}}}_{\text{residual component}}$$

$$= \hat{T}_{kj} + \tilde{T}_{kj} \tag{11}$$

### 2.4 The indicator of fault analysis

**2.4.1 Squared Prediction Error (SPE).** The Squared Prediction Error (SPE) is a metric employed for the detection of defects within residual space.

$$SPE = \tilde{T}_{kj}^T \tilde{T}_{kj} \tag{12}$$

The process is designed to identify abnormal operation or the presence of a fault, conditional upon the verification of the following condition [54].

$$SPE > \delta_\alpha^2 = \theta_1 \left[ 1 + \frac{C_\alpha \sqrt{2\theta_2 h_0^2}}{\theta_1} + \frac{\theta_2 h_0 (h_0 - 1)}{\theta_1^2} \right]^{h_0^{-1}} \tag{13}$$

In this particular instance, the symbol $\delta_\alpha^2$ represents the *SPE* detection threshold for the confidence threshold $\alpha$ reduced to the frequency domain.

$$\theta_i = \sum_{j=l+1}^{m} \lambda_j^i, \quad i = 1, 2, 3 \tag{14}$$

Where $\lambda_j^i$ is the $j^{th}$ eigenvalue of $\lambda$ raised to the power of *i*.

$$h_0 = 1 - \frac{2\theta_1 \theta_3}{3\theta_2^2} \tag{15}$$

and

$$C_\alpha = \frac{\theta_1}{\sqrt{2\theta_2 h_0^2}} \left[ \left( \frac{\|\tilde{T}_{kj}\|^2}{\theta_1} \right)^{h_0} - \left( \frac{\theta_2 h_0 (h_0 - 1)}{\theta_1^2} + 1 \right) \right] \tag{16}$$

**2.4.2 Hotelling $T^2$ index.** Hotelling's $T^2$ index is a measure of variations in the projections of observations in main space. It is determined from the first principal components obtained by the FFT in the frequency domain [55].

$$T^2 = \hat{T}_{kj} \hat{\Lambda}^{-1} \hat{T}_{kj} \tag{17}$$

$$\Lambda = \begin{bmatrix} \Lambda^{(l)} & \mathbf{0} \\ \mathbf{0} & \Lambda^{(m-l)} \end{bmatrix} = [\hat{\Lambda}|\tilde{\Lambda}] \tag{18}$$

$\hat{\Lambda}$ is a diagonal matrix containing the first $l$ eigenvalues of the correlation matrix in the frequency domain. The process is assumed to be an anomalous deviation if the following condition is valid:

$$T^2 > \chi_\alpha^2 = \frac{(N-1)(N+1)}{N(N-l)} F_{l,(N-l),\alpha} \sim \frac{N(N-l)}{l(N^2-1)} T^2 \tag{19}$$

where $\chi_\alpha^2$ is the confidence detection threshold associated with $\alpha$ and $F_{l,(N-l),\alpha}$ is the Fisher distribution.

**2.4.3 Kernel Principal Component Analysis (KPCA) Application.** The objective of KPCA is to address the limitations of the conventional PCA algorithm, particularly its inability to effectively capture nonlinear structures in data. The method involves projecting data from a lower-dimensional input space into a higher-dimensional feature space using a nonlinear mapping, via a kernel function, while ensuring a zero-mean condition in the new space [56]. In this transformed space, the classical PCA framework remains applicable. KPCA thus enables the extraction of more informative principal components when the data structure is inherently nonlinear.

Let $\Phi$ be the projection matrix of the input data matrix $X$ in the new feature space. The projection function is defined as follows:

$$\phi : x_i \in \mathbb{R}^m \longrightarrow \phi_i = \phi(x_i) \in \mathbb{R}^h, \quad \text{with} \quad m < h,$$

where $\phi$ is a nonlinear mapping from the input space to a higher-dimensional feature space.

The projected data matrix is given by:

$$\Phi = \begin{bmatrix} \phi(x_1) & \phi(x_2) & \phi(x_3) & \cdots & \phi(x_n) \end{bmatrix}. \tag{20}$$

To perform KPCA, it is necessary to center the projected data in the feature space. This leads to the zero-mean condition:

$$\frac{1}{N} \sum_{i=1}^{N} \phi(x_i) = 0. \tag{21}$$

The kernel function employed is the Gaussian (RBF) kernel, defined as:

$$k(x_i, x_j) = \exp\left(\mu \|x_i - x_j\|^2\right), \tag{22}$$

with

$$\mu = -\frac{1}{2\tau^2},$$

where $\tau$ denotes the bandwidth of the Gaussian kernel.

## 3 Results and discussion

The method developed in this article has been validated using experimental data obtained at the Bearings Data Center at Case Western Reserve University [57]. As shown in Fig 3 and Table 1, the test rig consists of an electrical motor, an accelerometer and a torque encoder/transducer. The test bearing is an SKF 6205-2RS ball bearing. The bearing inner race, outer race and rolling element were each subjected to artificial seeding of a single point fault, achieved by means of electro-discharge machining. This methodical approach enabled a rigorous evaluation of the bearings under various operating conditions. Vibration data were collected using special magnets and sensors attached to the motor housing. For this

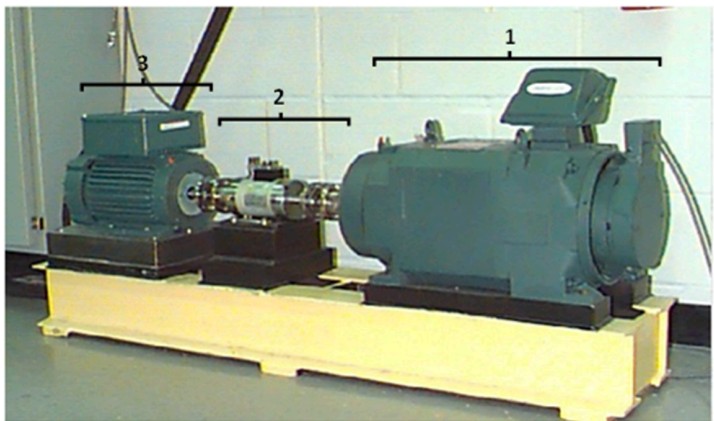

**Fig 3**. Experimental bench for collecting vibration data from bearings (CWRU) [57].

**Table 1**. Dimensional characteristics of the bearing.

| Inside diameter (mm) | Outside diameter (mm) | Thickness (mm) | Ball diameter (mm) | Pitch diameter (mm) | Number of ball rolling |
|---|---|---|---|---|---|
| 25 | 52 | 15 | 8 | 40 | 9 |

study, vibration signals from the driven end (DE) of the bearing were recorded at a sampling frequency of 12 kHz. In addition to normal operating conditions, the database includes three types of vibration signal defects located on the outer ring, inner ring, and bearing ball, respectively. The widths of the defects identified as T1, T2, and T3 are 0.1778 mm, 0.3556 mm, and 0.5334 mm, respectively; the depth common to all three defects is 0.2794 mm. Measurement data was recorded at different rotational speeds: 1,730, 1,750, 1,772, and 1,797 rpm.This methodical approach enabled a rigorous evaluation of the bearings under various operating conditions . The results are presented in three sub-sections: one covering the defects in the outer ring, the other those in the inner ring and the last in ball.

### 3.1 Ball bearing outer ring

The results of ball bearings for outer ring defects are illustrated in Figs 4, 5, 6 and 7. Two conventional methods are employed for the analysis of the defects, utilising the indices $T^2$ for the principal component and *SPE* for the residuals. Subsequently, both methods are combined with a Fast Fourier Transform (FFT). The results demonstrate the presence of the fault in accordance with the statistical indicators $T^2$ and SPE. It is evident that in the healthy state (blue curve), the defect indicators (red dotted curve) exceed the test data. While conventional methods are effective in identifying defects, their interpretation can be challenging in certain instances, as illustrated in Figs 4(b)-i, 5(b)-i, 6(b)-i and 7(b)-i. The combination of these methods with the FFT provides a more definitive indication of the fault. Figs 4(c), 4(d), 5(c), 5(d), 6(c), 6(d), 7(c) and 7(d) provide clear evidence of the presence of defects. In contrast to Figs 4(a), 4(b), 5(a), 5(b), 6(a), 6(b), 7(a) and 7(b). The $T^2$ indicator is most effective when used in conjunction with PCA-FFT and SPE for KPCA-FFT. When employed in an optimal manner, these indicators yield highly accurate detection results.

### 3.2 Ball bearing inner ring

The results obtained on the inner ring of the ball bearing are presented in Figs 8, 9, 10 and 11. As previously observed, the combination of conventional analysis methods with FFT yields highly conclusive results, effectively identifying the fault with greater precision. The SPE and $T^2$ performance indicators are clearly established, and a discrepancy is evident between healthy data and data exhibiting faults.

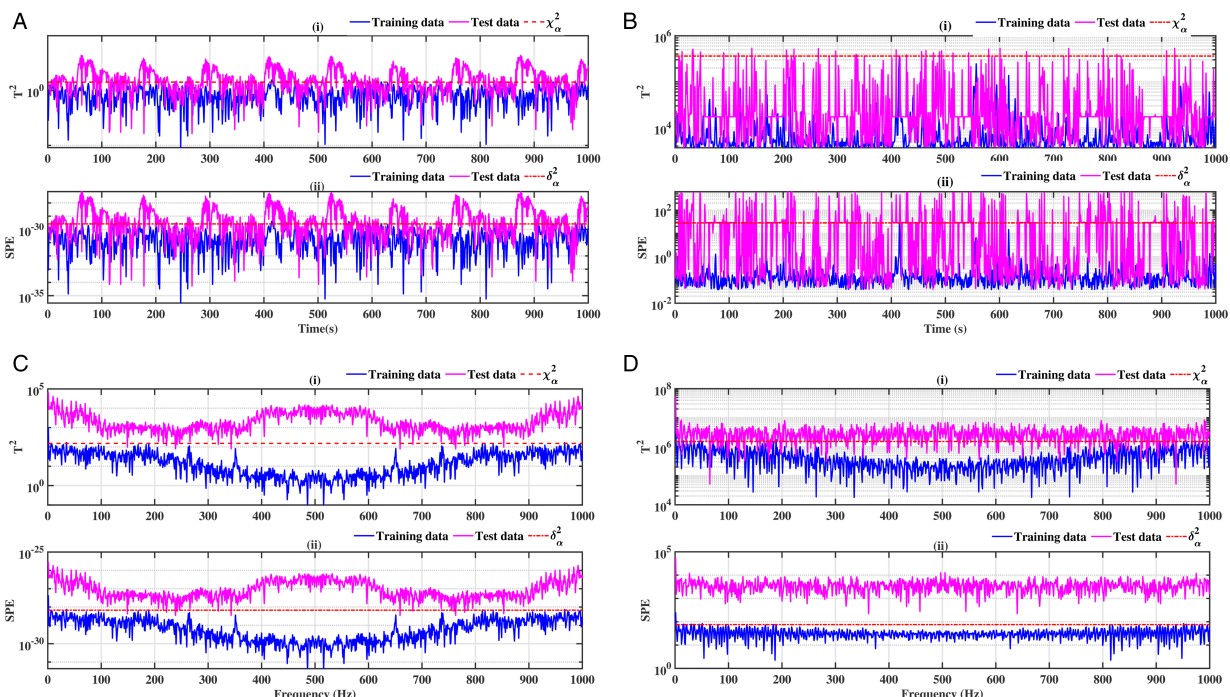

**Fig 4**. Fault on size T1 outer ring at a speed of 1730 rpm: a-i), b-i), c-i) and d-i) analysis by $T^2$ indicator; a-ii), b-ii), c-ii) and d-ii) analysis by SPE indicator.

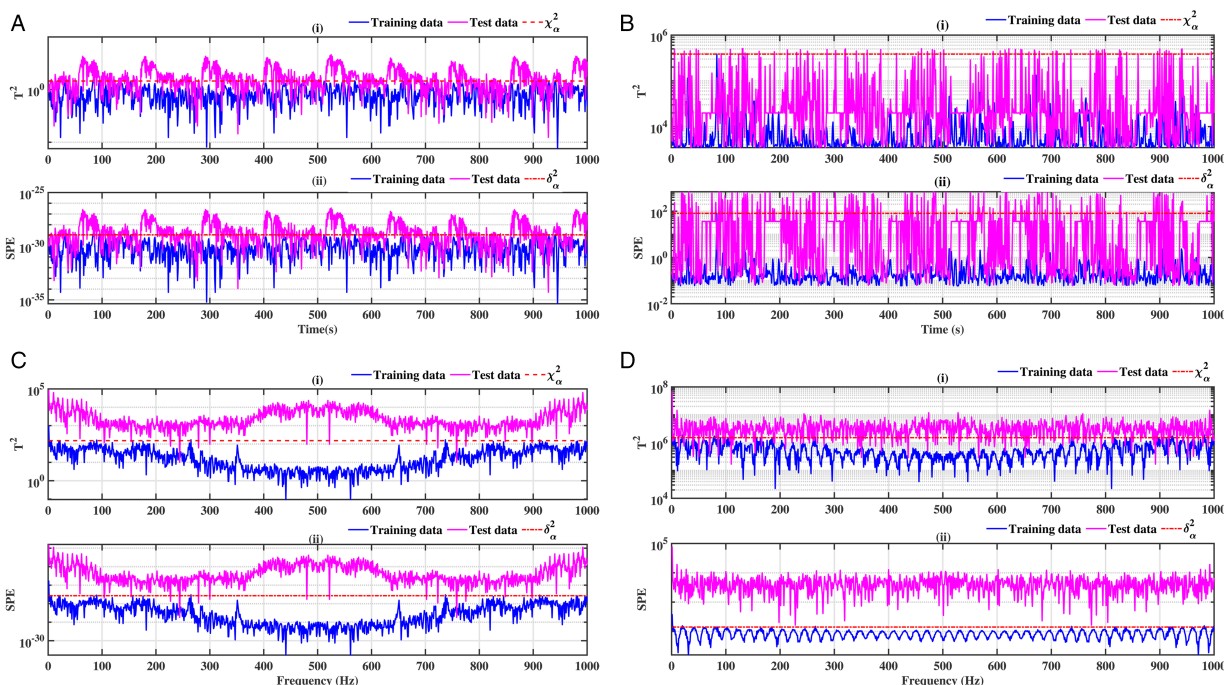

**Fig 5**. Fault on size T1 outer ring at a speed of 1750 rpm: a-i), b-i), c-i), and d-i) analysis by $T^2$ indicator; a-ii), b-ii), c-ii), and d-ii) analysis by SPE indicator.

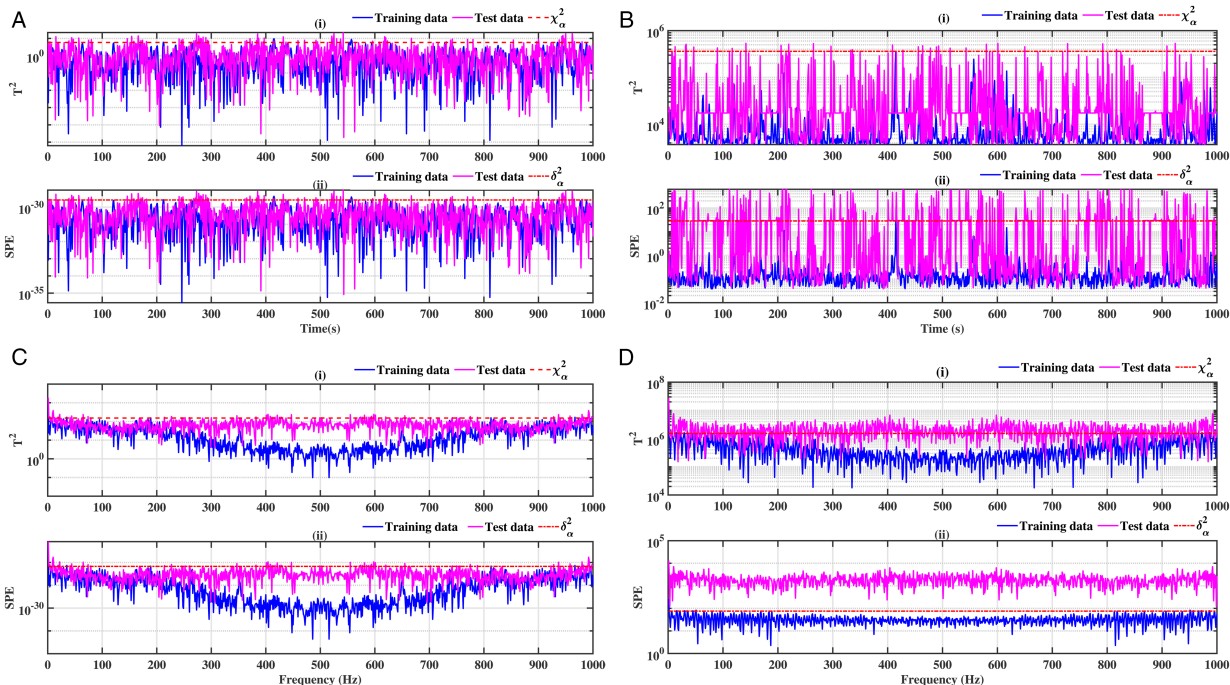

**Fig 6. Fault on size T2 outer ring at a speed of 1730 rpm: a-i), b-i), c-i), and d-i) analysis by $T^2$ indicator; a-ii), b-ii), c-ii), and d-ii) analysis by SPE indicator.**

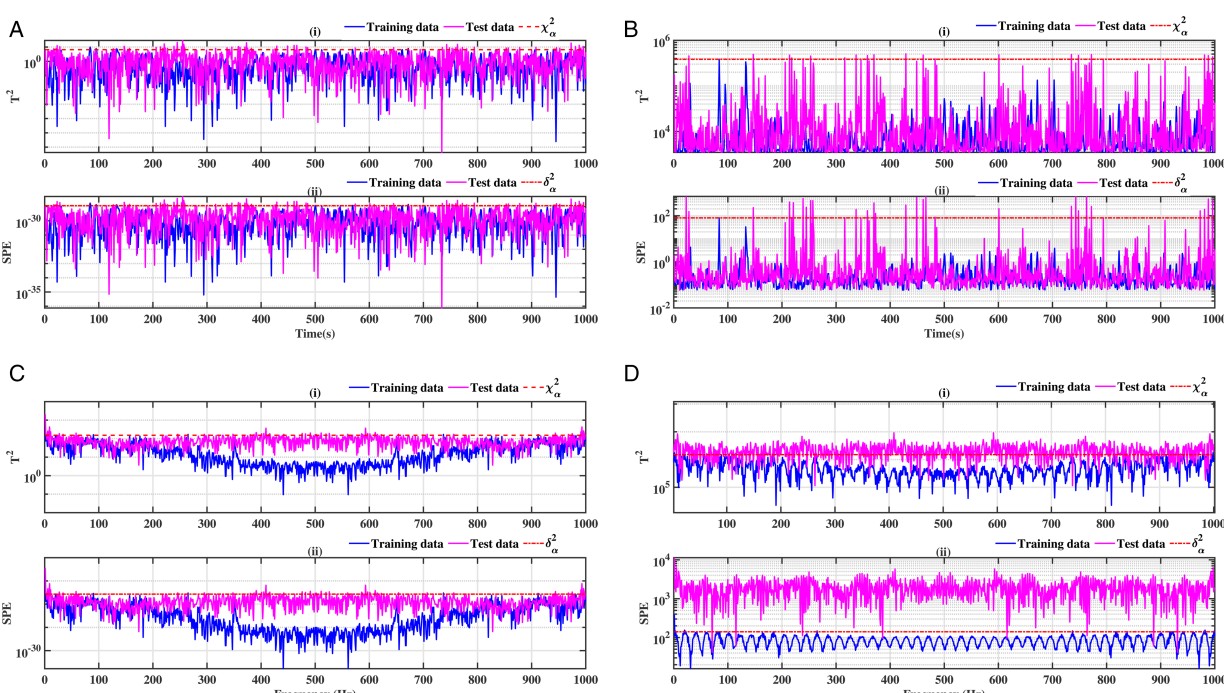

**Fig 7. Fault on size T2 outer ring at a speed of 1750 rpm:a-i), b-i), c-i), and d-i) analysis by $T^2$ indicator; a-ii), b-ii), c-ii), and d-ii) analysis by SPE indicator.**

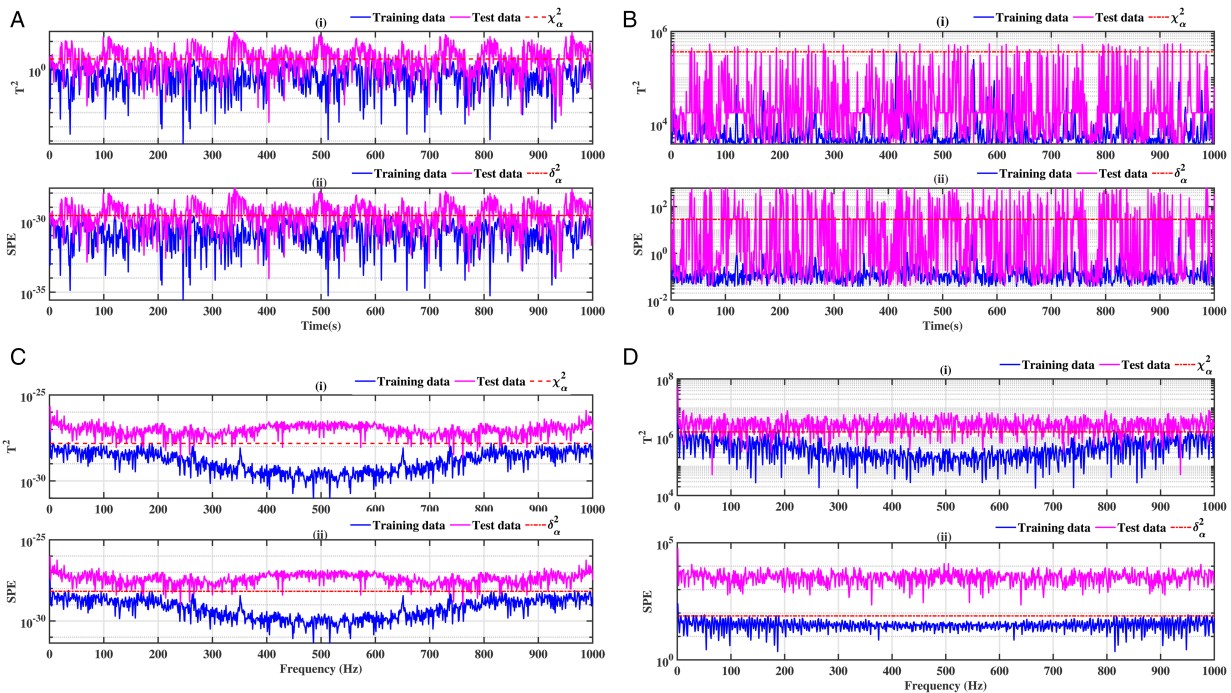

**Fig 8**. Fault on size T1 inner ring at a speed of 1730 rpm: a-i), b-i), c-i), and d-i) analysis by $T^2$ indicator; a-ii), b-ii), c-ii), and d-ii) analysis by SPE indicator.

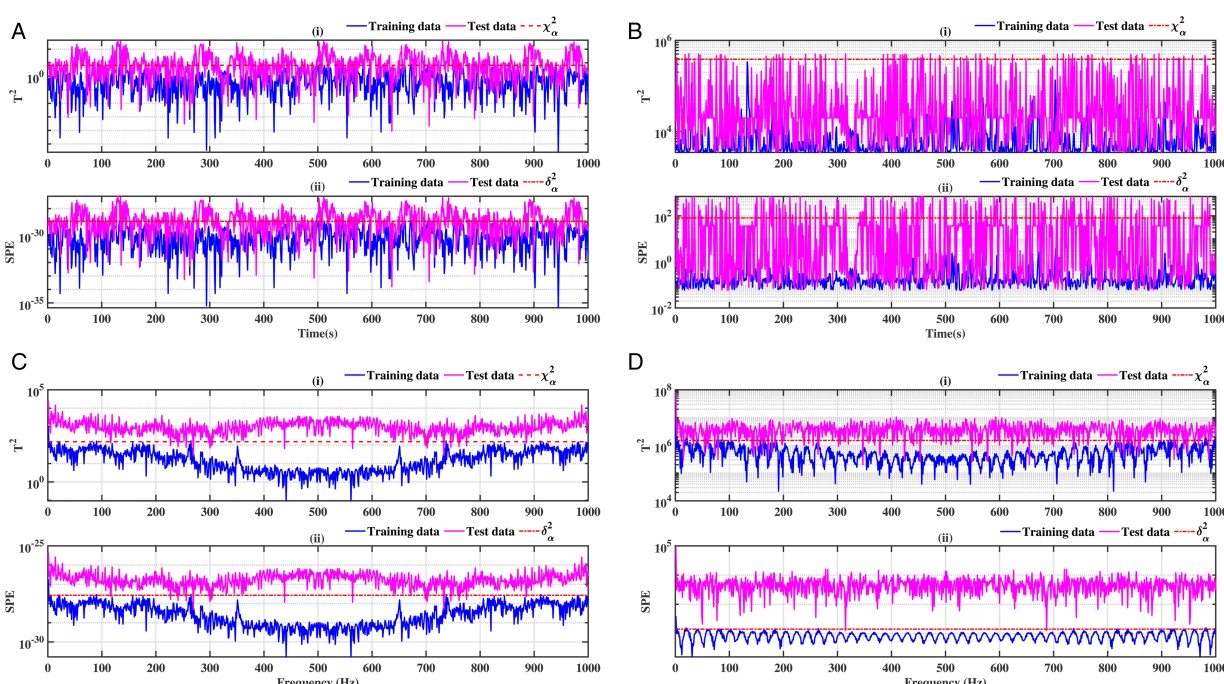

**Fig 9**. Fault on size T1 inner ring at a speed of 1750 rpm: a-i), b-i), c-i), and d-i) analysis by $T^2$ indicator; a-ii), b-ii), c-ii), and d-ii) analysis by SPE indicator.

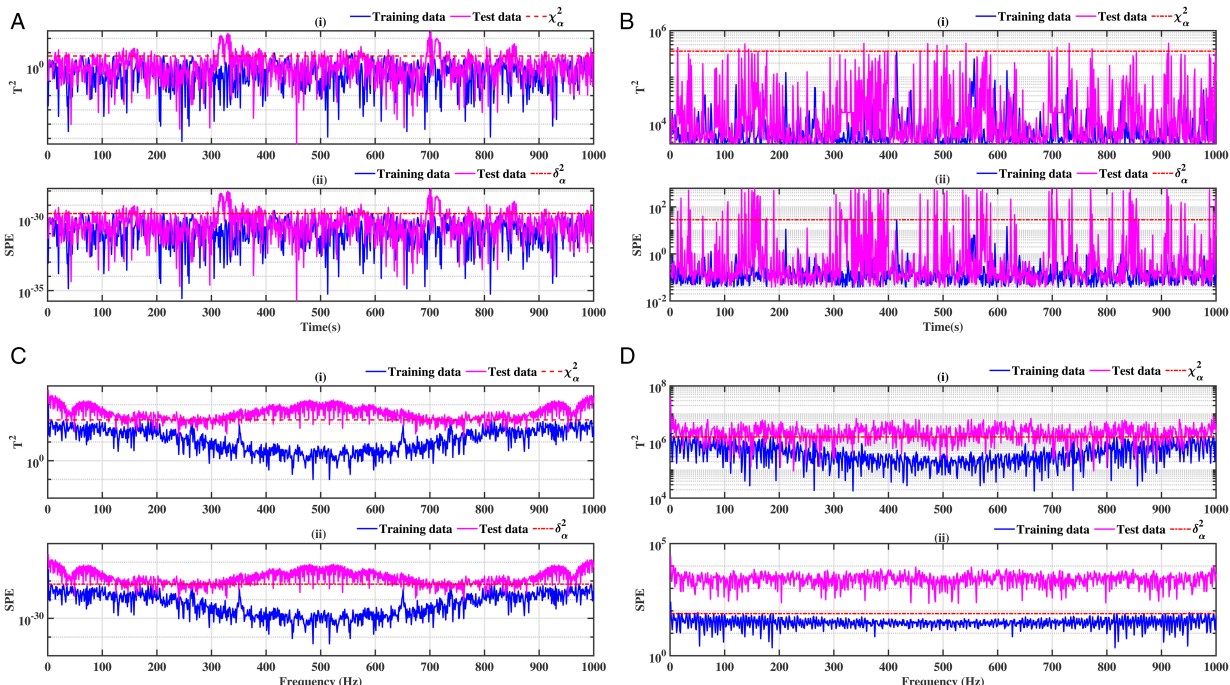

**Fig 10. Fault on size T2 inner ring at a speed of 1730 rpm: a-i), b-i), c-i), and d-i) analysis by $T^2$ indicator; a-ii), b-ii), c-ii), and d-ii) analysis by SPE indicator.**

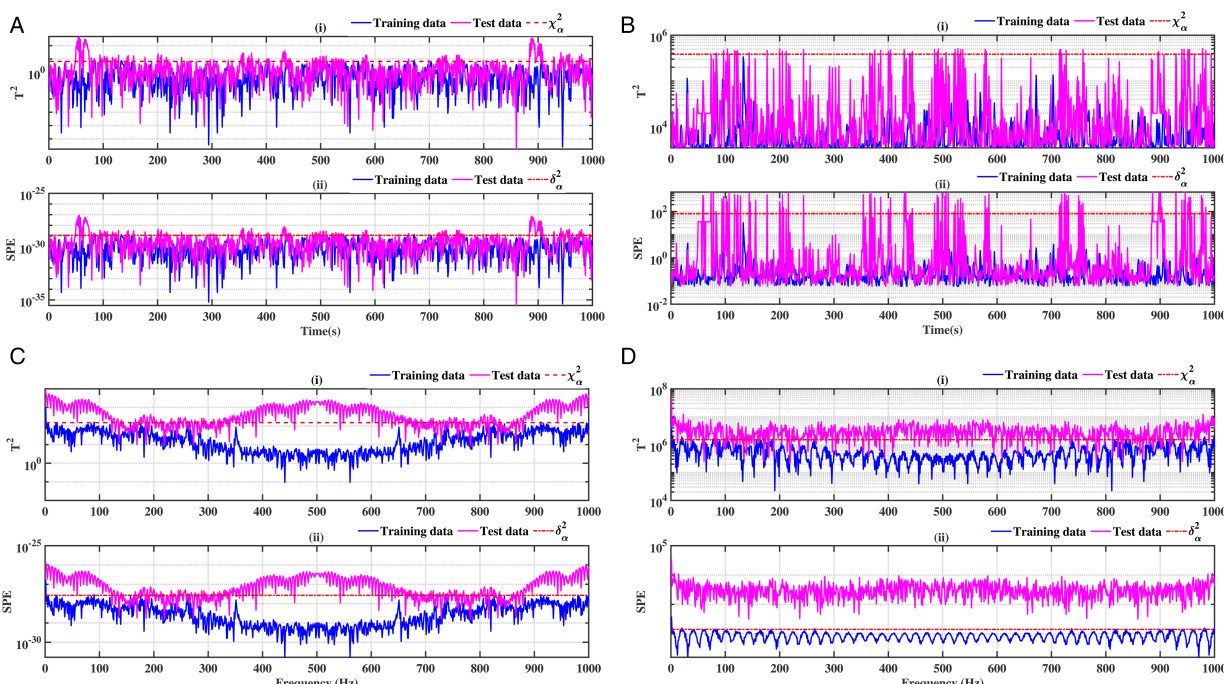

**Fig 11. Fault on size T2 inner ring at a speed of 1750 rpm: a-i), b-i), c-i), and d-i) analysis by $T^2$ indicator; a-ii), b-ii), c-ii), and d-ii) analysis by SPE indicator.**

## 3.3 Ball bearing elements rolling

The results relating to the rolling elements of the ball bearing, illustrated in Figs 12, 13, 14 and 15, confirm the relevance of the proposed methods. Indeed, the integration of conventional analysis techniques, such as PCA and KPCA, with the Fast Fourier Transform, substantially improves defect detection. This combination has been demonstrated to enhance the readability of statistical indicators such as Hotelling's $T^2$ and SPE. It is evident that the presence of anomalies is indicated by the exceeding of the threshold of these indicators. It has been demonstrated that the PCA-FFT and KPCA-FFT methods offer a more marked separation between healthy and defective signals, resulting in improved sensitivity and reliability of diagnosis

## 3.4 Comparative evaluation

The evaluation of the dispersion of signals between healthy and faulty systems, as a function of fault sizes, at various speeds, can be obtained by means of the calculation of the Symmetric Mean Absolute Percentage Error (SMAPE) [58].

$$SMAPE = \frac{1}{n} \sum_{i=1}^{n} \frac{2|y_i - \hat{y}_i|}{|y_i| + |\hat{y}_i|} \tag{23}$$

In this equation, $n$ denotes the width of the signal, $y_i$ denotes the predicted vector, and $y_i$ denotes the actual vector. Tables 2, 3 and Figs 16, 17, 18, and 19 present the findings of calculations of dispersions, in both principal and residual

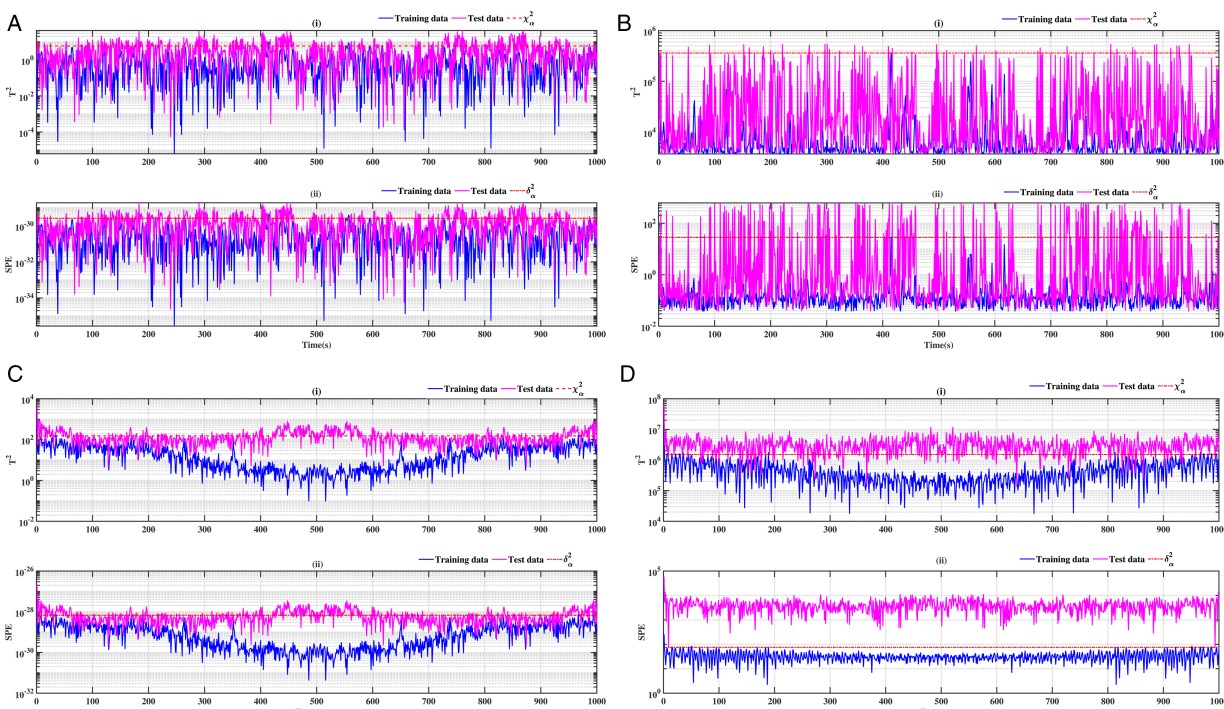

**Fig 12. Fault on size T1 ball at a speed of 1730 rpm: a-i), b-i), c-i), and d-i) analysis by $T^2$ indicator; a-ii), b-ii), c-ii), and d-ii) analysis by SPE indicator.**

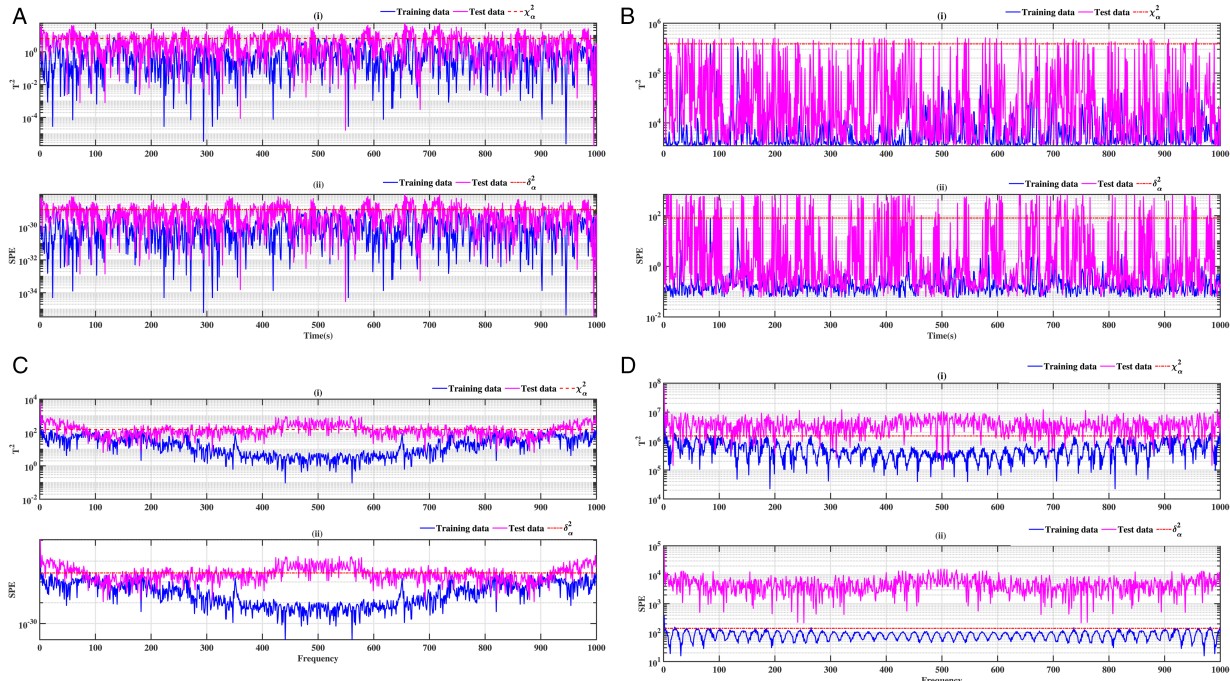

**Fig 13. Fault on size T1 ball at a speed of 1750 rpm: a-i), b-i), c-i), and d-i) analysis by $T^2$ indicator; a-ii), b-ii), c-ii), and d-ii) analysis by SPE indicator.**

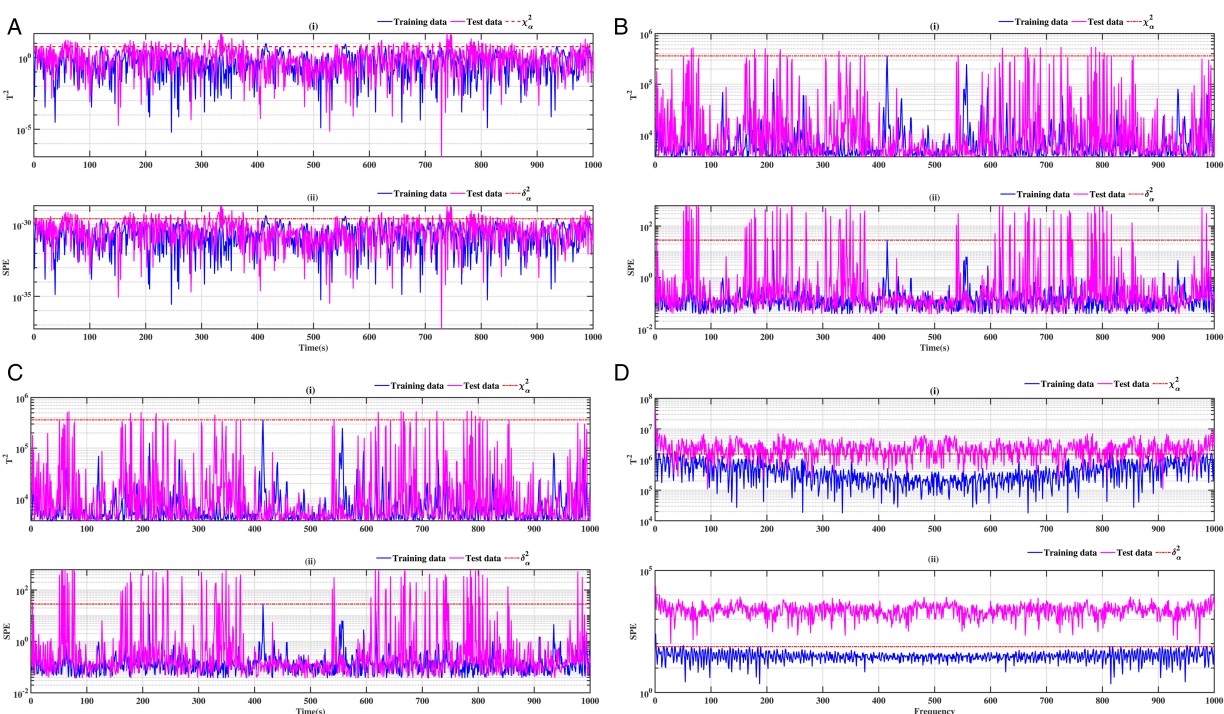

**Fig 14. Fault on size T2 ball at a speed of 1730 rpm: a-i), b-i), c-i), and d-i) analysis by $T^2$ indicator; a-ii), b-ii), c-ii), and d-ii) analysis by SPE indicator.**

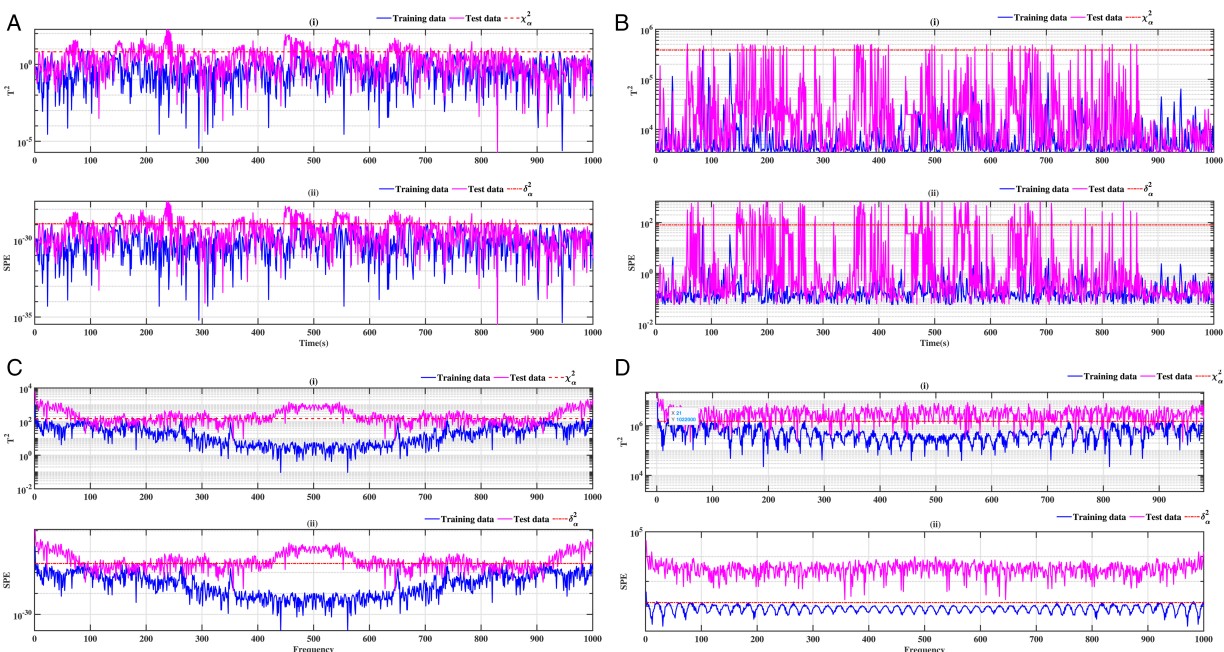

**Fig 15**. Fault on size T2 ball at a speed of 1750 rpm: a-i), b-i), c-i), and d-i) analysis by $T^2$ indicator; a-ii), b-ii), c-ii), and d-ii) analysis by SPE indicator.

**Table 2**. Dispersion signal in principal and residual space in PCA and KPCA algorithm.

| Methods | | Speed (Rpm) | T1 | Dispersion T2 | T3 | Healthy | T1 | Dispersion T2 | T3 | Healthy |
|---------|---|-------------|-----|---------------|-----|---------|-----|---------------|-----|---------|
| | | | $T^2$ | | | | SPE | | | |
| PCA | Inner ring | 1730 | 0.0015 | 0.0013 | 0.0016 | 0.0013 | 0.0015 | 0.0013 | 0.0016 | 0.0013 |
| | | 1750 | 0.0016 | 0.0014 | 0.0017 | | 0.0016 | 0.0014 | 0.0017 | |
| | | 1772 | 0.0015 | 0.0013 | 0.0016 | | 0.0015 | 0.0013 | 0.0016 | |
| | | 1797 | 0.0015 | 0.0013 | 0.0016 | | 0.0015 | 0.0013 | 0.0016 | |
| | Outer ring | 1730 | 0.0015 | 0.0013 | 0.0015 | | 0.0015 | 0.0013 | 0.0015 | |
| | | 1750 | 0.0016 | 0.0013 | 0.0015 | | 0.0016 | 0.0013 | 0.0015 | |
| | | 1772 | 0.0016 | 0.0013 | 0.0015 | | 0.0016 | 0.0013 | 0.0015 | |
| | | 1797 | 0.0016 | 0.0013 | 0.0015 | | 0.0016 | 0.0013 | 0.0015 | |
| | Ball | 1730 | 0.0014 | 0.0013 | 0.0013 | | 0.0014 | 0.0013 | 0.0013 | |
| | | 1750 | 0.0014 | 0.0014 | 0.0013 | | 0.0014 | 0.0014 | 0.0013 | |
| | | 1772 | 0.0014 | 0.0014 | 0.0015 | | 0.0014 | 0.0014 | 0.0015 | |
| | | 1797 | 0.0014 | 0.0014 | 0.0013 | | 0.0014 | 0.0014 | 0.0013 | |
| KPCA | Inner ring | 1730 | 0.001 | 7.97E-04 | 0.001 | 4.72E-04 | 0.0014 | 9.95E-04 | 0.0015 | 6.97E-04 |
| | | 1750 | 0.0011 | 8.34E-04 | 0,0011 | 4.98E-04 | 0.0014 | 9.79E-04 | 0.0016 | 7.00E-04 |
| | | 1772 | 0.001 | 7.61E-04 | 0.001 | 5.42E-04 | 0.0014 | 9.18E-04 | 0.0014 | 7.34E-04 |
| | | 1797 | 9.98E-04 | 7.67E-04 | 0.001 | 5.94E-04 | 0.0013 | 9.79E-04 | 0.0015 | 7.27E-04 |
| | Outer ring | 1730 | 0.001 | 6.57E-04 | 8.61E-04 | 4.72E-04 | 0.0014 | 8.34E-04 | 0.0012 | 6.97E-04 |
| | | 1750 | 0.0011 | 7.59E-04 | 9.72E-04 | 4.98E-04 | 0.0015 | 8.55E-04 | 0.0012 | 7.00E-04 |
| | | 1772 | 0.0011 | 7.73E-04 | 9.31E-04 | 5.42E-04 | 0.0015 | 8.86E-04 | 0.0012 | 7.34E-04 |
| | | 1797 | 0.0011 | 6.69E -04 | 9.02E-04 | 5.94E-04 | 0.0015 | 8.36E-04 | 0.0012 | 7.27E-04 |
| | Ball | 1730 | 9.60E-04 | 7.07E-04 | 8.13E-04 | 4.72E-04 | 0.0011 | 8.94E-04 | 9.98E-04 | 6.97E-04 |
| | | 1750 | 0.001 | 9.36E-04 | 8.41E-04 | 4.98E-04 | 0.0012 | 0.0011 | 9.40E-04 | 7.00E-04 |
| | | 1772 | 9.40E-04 | 8.22E-04 | 0.001 | 5.42E-04 | 0.0011 | 0.0011 | 0.0012 | 7.34E-04 |
| | | 1797 | 8.79E-04 | 8.28E-04 | 6.86E-04 | 5.94E-04 | 0.001 | 9.93E-04 | 8.86E-04 | 7.27E-04 |

**Table 3**. Dispersion signal in principal and residual space in PCA-FFT and KPCA-FFT algorithm.

| Methods | | Speed (Rpm) | T1 | Dispersion T2 | T3 | Healthy | T1 | Dispersion T2 | T3 | Healthy |
|---|---|---|---|---|---|---|---|---|---|---|
| | | | $T^2$ | | | | SPE | | | |
| PCA-FFT | Inner ring | 1730 | 0.0018 | 0.0017 | 0.0019 | 6.94E-04 | 0.0018 | 0.0017 | 0.0019 | 0.0012 |
| | | 1750 | 0.0019 | 0.0016 | 0.0019 | | 0.0019 | 0.0016 | 0.0019 | 0.0014 |
| | | 1772 | 0.0018 | 0.0018 | 0.0019 | | 0.0018 | 0.0019 | 0.0019 | 0.0015 |
| | | 1797 | 0.0018 | 0.0018 | 0.0019 | | 0.0018 | 0.0014 | 0.0019 | 0.0016 |
| | Outer ring | 1730 | 0.0019 | 0.0012 | 0.002 | | 0.0019 | 0.0012 | 0.002 | 0.0012 |
| | | 1750 | 0.0019 | 0.0012 | 0.002 | | 0.0019 | 0.0012 | 0.002 | 0.0014 |
| | | 1772 | 0.0019 | 0.0012 | 0.0019 | | 0.0019 | 0.0012 | 0.002 | 0.0015 |
| | | 1797 | 0.0019 | 0.0012 | 0.002 | | 0.0019 | 0.0012 | 0.002 | 0.0016 |
| | Ball | 1730 | 0.0014 | 0.0013 | 0.0013 | | 0.0014 | 0.0013 | 0.0013 | 0.0012 |
| | | 1750 | 0.0014 | 0.0015 | 0.0013 | | 0.0014 | 0.0015 | 0.0013 | 0.0014 |
| | | 1772 | 0.0014 | 0.0015 | 0.0017 | | 0.0014 | 0.0015 | 0.0017 | 0.0015 |
| | | 1797 | 0.0013 | 0.0015 | 0.0013 | | 0.0013 | 0.0015 | 0.0013 | 0.0016 |
| KPCA-FFT | Inner ring | 1730 | 0.0015 | 0.0013 | 0.0014 | 6.0754E-04 | 0.002 | 0.0019 | 0.002 | 0.0012 |
| | | 1750 | 0.0014 | 0.0013 | 0.0013 | 6.2953E-04 | 0.0019 | 0.0019 | 0.0019 | 0.0014 |
| | | 1772 | 0.0013 | 0.0012 | 0.0014 | 7.3429E-04 | 0.0019 | 0.0019 | 0.0019 | 0.0015 |
| | | 1797 | 0.0015 | 0.0013 | 0.0014 | 8.6839E-04 | 0.002 | 0.0019 | 0.0019 | 0.0016 |
| | Outer ring | 1730 | 0.0014 | 0.0012 | 0.0013 | 6.0754E-04 | 0.002 | 0.0019 | 0.0019 | 0.0012 |
| | | 1750 | 0.0014 | 0.0011 | 0.0013 | 6.2953E-04 | 0.0019 | 0.0018 | 0.0019 | 0.0014 |
| | | 1772 | 0.0014 | 0.0012 | 0.0013 | 7.3429E-04 | 0.0019 | 0.0018 | 0.0019 | 0.0015 |
| | | 1797 | 0.0014 | 0.0012 | 0.0014 | 8.6839E-04 | 0.002 | 0.0019 | 0.0019 | 0.0016 |
| | Ball | 1730 | 0.0014 | 0.0013 | 0.0014 | 6.0754E-04 | 0.002 | 0.0019 | 0.0019 | 0.0012 |
| | | 1750 | 0.0014 | 0.0013 | 0.0013 | 6.2953E-04 | 0.0019 | 0.0019 | 0.0018 | 0.0014 |
| | | 1772 | 0.0013 | 0.0013 | 0.0014 | 7.3429E-04 | 0.0019 | 0.0019 | 0.0019 | 0.0015 |
| | | 1797 | 0.0014 | 0.0014 | 0.0013 | 8.6839E-04 | 0.0019 | 0.002 | 0.0019 | 0.0016 |

space, between healthy and defective signal components of the bearing components. In the main plane, the resulting dispersions of the healthy and faulty signals are significantly larger than those of the healthy signals at variable speeds. For inner ring faults, the values range from 0.0012 to 0.0015, for outer ring faults they range from 0.0011 to 0.0014, and for ball faults they range from 0.0012 to 0.0014 when using KPCA-FFT. In the case of PCA-FFT, the dispersion varies from 0.0016 to 0.0018 for inner ring defects, 0.0012 to 0.002 for outer ring defects and 0.0012 to 0.0017 for ball defects. The detection of defects in the residual space, as ensured by the SPE, demonstrates that the KPCA-FFT model is more effective in detecting these defects, regardless of their position, with a dispersion oscillating between 0.0017 and 0.002. Conversely, the PCA-FFT model detects defects in the inner ring with a dispersion ranging from 0.0016 to 0.0018. Defects of size 0.355 mm (T2) on the outer ring are undetectable at speeds exceeding 1730 rpm. At speeds ranging from 1738 rpm to 1755 rpm and at speeds above 1788 rpm, defects of size T3 are undetectable. Finally, at speeds in excess of 1770 rpm, defects of size T2 are undetectable. Finally, at speeds in excess of 1750 rpm, the T1 defect is undetectable. Consequently, at speeds beyond 1770 rpm, the T2 defect becomes undetectable, and at speeds exceeding 1750 rpm, the T1 defect becomes undetectable.

## 3.5 Synthesis

A comparative analysis of the results indicates that the proposed methods are more effective than the conventional PCA and KPCA methods in identifying faults.The primary innovation of this study lies in the integration of projection techniques (PCA and KPCA) with frequency analysis (FFT), combined with the simultaneous use of Hotelling's $T^2$ and SPE statistical indices for anomaly identification. This hybrid approach enables a dual interpretation of vibration signals, both in the principal space ($T^2$) and in the residual space ($SPE$). It has been demonstrated that the KPCA+FFT model

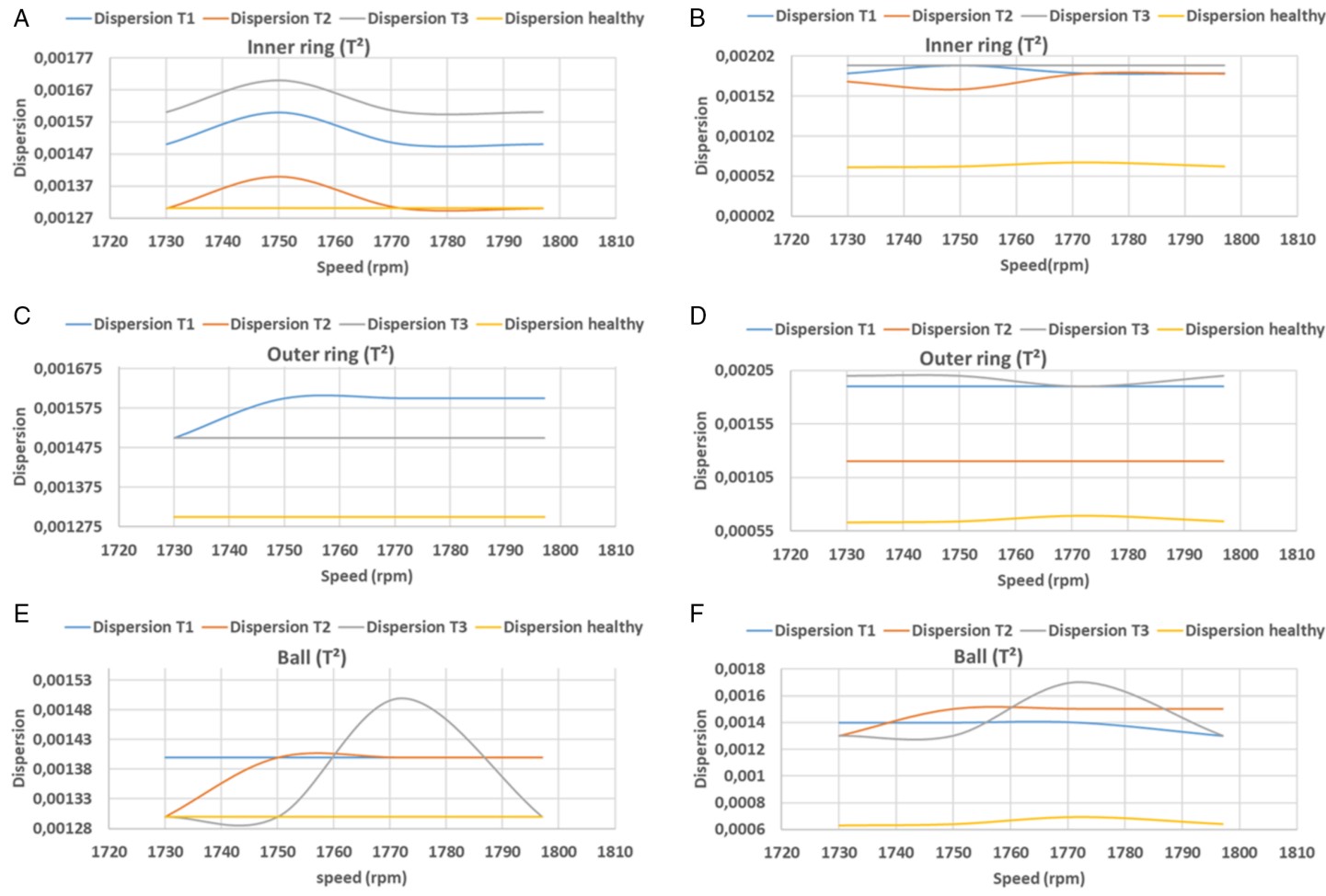

**Fig 16**. PCA and PCA-FFT dispersion curves for $T^2$ index test defect sizes.

provides enhanced discrimination capacity in the residual space, particularly for defects characterised by low amplitude or masked by noise (see Figs 4(d)-ii, 5(d)-ii, 6(d)-ii, 7(d)-ii, 8(d)-ii, 9(d)-ii, 10(d)-ii, 11(d)-ii, 12(d)-ii, 13(d)-ii, 14(d)-ii, 15(d)-ii and Table 3 with a dispersion in the interval [0.0017; 0.002]. This observation is supported by experimental vibration data of fault rolling bearings. However, it has been observed that the PCA+FFT model produces more satisfactory results in the principal space, indicating increased performance when defects are adequately represented in the first principal components (see Figs 4(c)-ii, 5(c)-ii, 6(c)-ii, 7(c)-ii, 8(c)-ii, 9(c)-ii, 10(c)-ii, 11(c)-ii, 12(c)-ii, 13(c)-ii, 14(c)-ii, 15(c)-ii), for dispersion ranges of [0.0016; 0.0018] for inner ring defects, [0.0012; 0.002] for outer ring defects, and [0.0012; 0.0017] for ball defects. This differentiation of performance based on the projection domain constitutes a significant methodological contribution. It suggests that a robust and reliable diagnosis could benefit from the combined use of both spaces, with the analysis being adapted to specific operating conditions or the type of defect suspected. This approach has been demonstrated to enhance the sensitivity and reliability of the diagnosis in comparison with conventional methods. The latter utilise either Principal Component Analysis (PCA) or Fast Fourier Transform (FFT) independently.

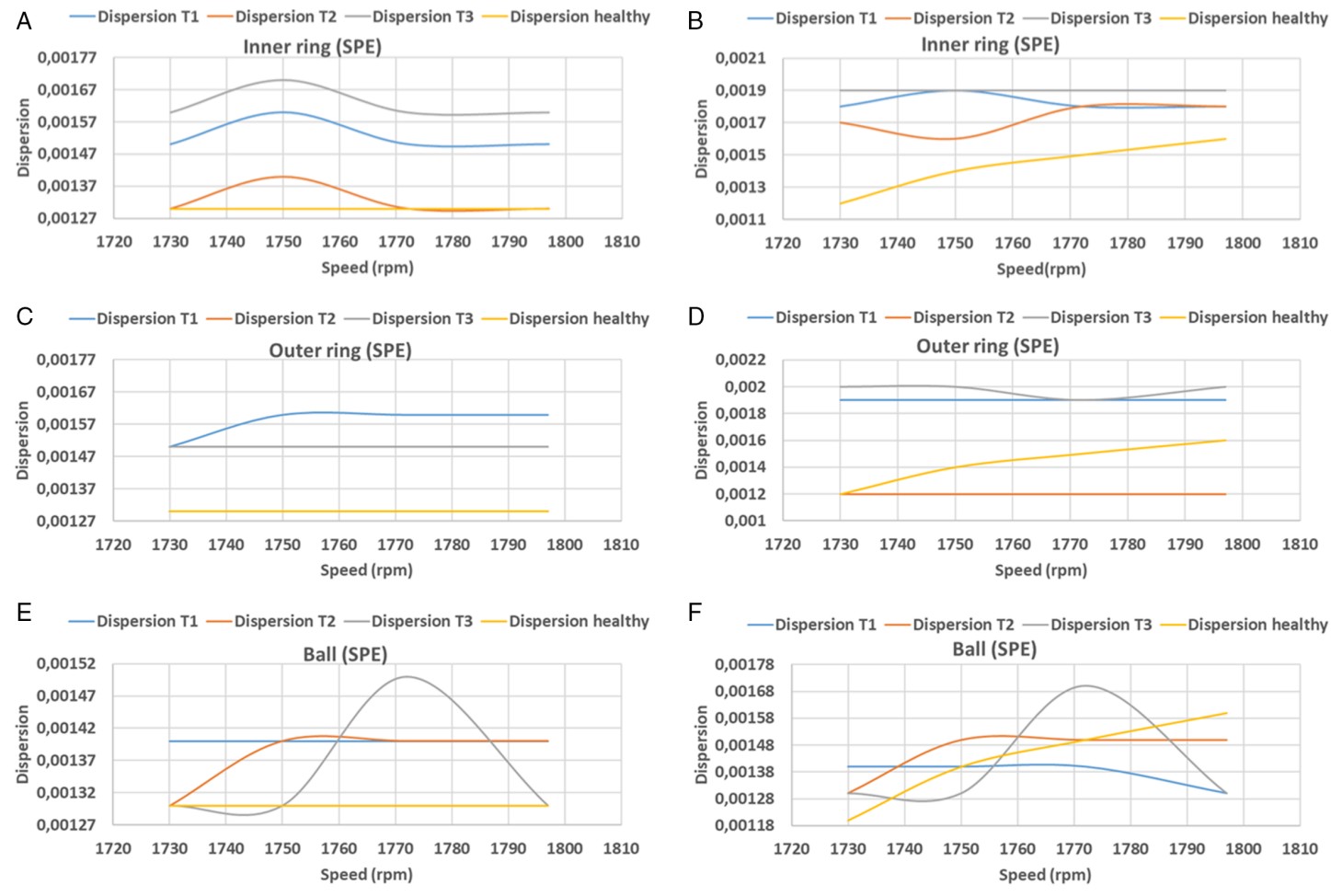

**Fig 17. PCA and PCA-FFT dispersion curves for *SPE* index test defect sizes.**

## 4 Conclusion

The objective of this paper is to present a methodology for the detection of bearing faults. The identified faults are located on the ball, outer and inner rings of the bearings. The fault detection results pertain to two sizes, T1 and T2, of the same depth, operating at speeds of 1730 rpm and 1,750 rpm, respectively. The diagnostic methods associated with the $T^2$ index test and SPE of the defects are presented and illustrated. It is demonstrated that all of the aforementioned methods are capable of detecting faults. The present study has demonstrated the efficacy of an integrated approach to diagnosing bearing faults, based on realistic modelling of a single-row rigid ball bearing commonly used in industrial applications. The judicious combination of structural filtering, FFT frequency analysis, and statistical indicators in the principal and residual subspaces resulted in enhanced discrimination between health states. The findings obtained provide substantial evidence for the efficacy of the PCA+FFT combination in the principal space for detecting vibration signals associated with defects. Conversely, the analysis undertaken with the KPCA+FFT combination applied to the residual space accentuates significant nonlinear components, which are frequently underutilised. These results confirm the value of a double reading of the data projected in the two subspaces resulting from dimensional reduction, thereby improving the

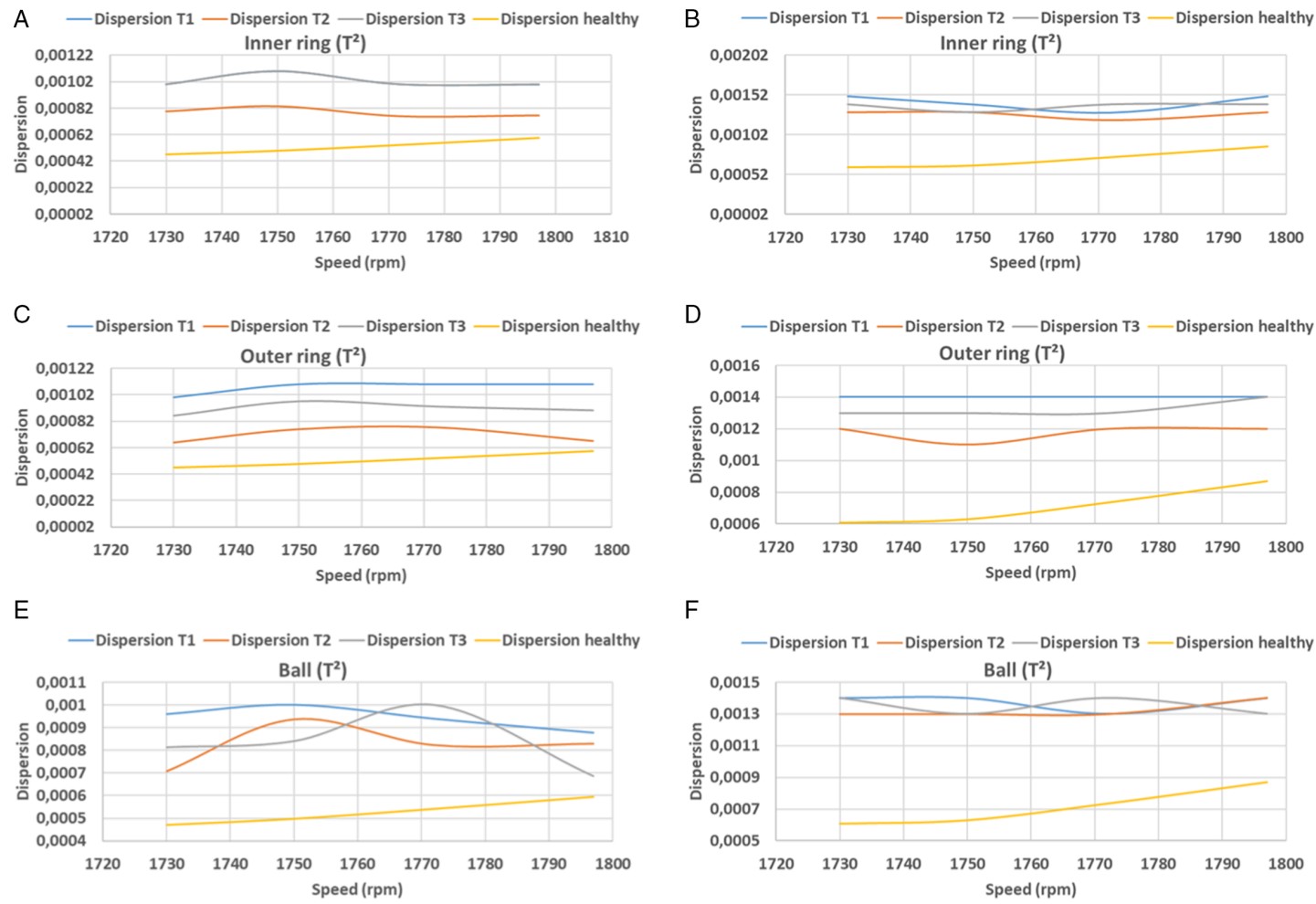

**Fig 18**. KPCA and KPCA-FFT dispersion curves for $T^2$ index test defect sizes.

overall diagnostic capability of the system. Despite the emphasis on a conventional architectural framework, the methodology devised exhibits considerable promise with regard to its generalisability. It is evident that the principles implemented are applicable to other types of bearings (roller bearings, angular contact bearings, etc.), subject to adaptation of the modelling and acquisition parameters. In view of these findings, future research endeavours will focus on extending this methodology to more complex mechanical architectures and a wider range of defect types, within realistic operating contexts.

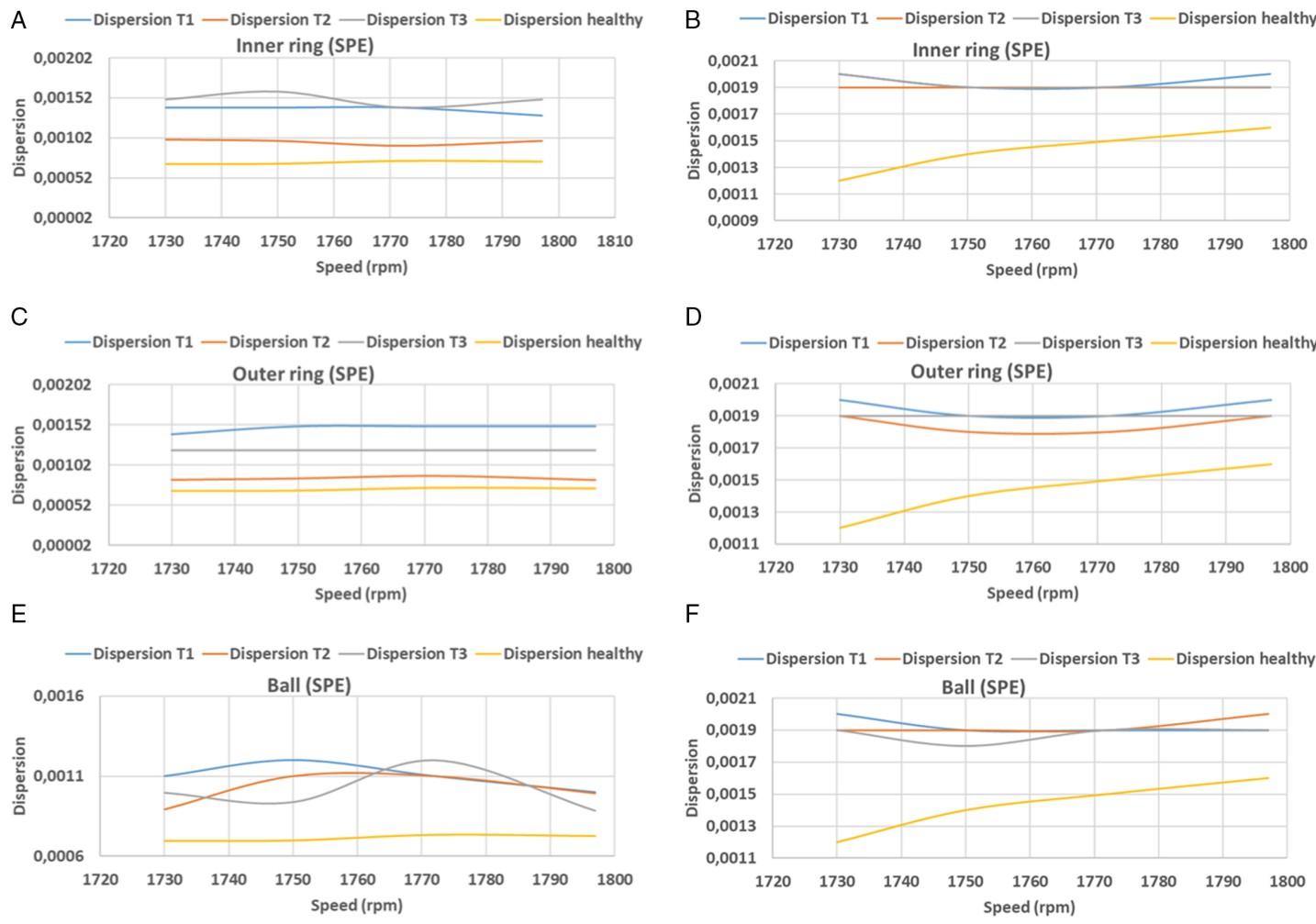

**Fig 19**. KPCA and KPCA-FFT dispersion curves for *SPE* index test defect sizes.

## Author contributions

**Data curation:** Emmanuel Ndongue Esseme.

**Formal analysis:** Emmanuel Tonye.

**Investigation:** Emmanuel Ndongue Esseme.

**Methodology:** Emmanuel Ndongue Esseme, Thomas Florent Noël Kanaa, Mathieu Jean Pierre Pesdjock.

**Software:** Thomas Florent Noël Kanaa, Thérèse Jacquie Ngo Bisse.

**Supervision:** Thomas Florent Noël Kanaa, Mathieu Jean Pierre Pesdjock, Emmanuel Tonye.

**Validation:** Emmanuel Ndongue Esseme, Thomas Florent Noël Kanaa, Thérèse Jacquie Ngo Bisse, Ludovic Ngongang, Emmanuel Tonye.

**Writing – original draft:** Emmanuel Ndongue Esseme, Thérèse Jacquie Ngo Bisse.

**Writing – review & editing:** Emmanuel Ndongue Esseme, Ludovic Ngongang.

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
