## [Decision Letter · Decision Letter 0]

17 Jun 2025

PONE-D-25-26658A simplified approach to failure analysis of ball bearings combining Principal Component Analysis and Fast Fourier TransformPLOS ONE

Dear Dr. Kanaa,

Thank you for submitting your manuscript to PLOS ONE. After careful consideration, we feel that it has merit but does not fully meet PLOS ONE’s publication criteria as it currently stands. Therefore, we invite you to submit a revised version of the manuscript that addresses the points raised during the review process.

We look forward to receiving your revised manuscript.

Kind regards,

Dandan Peng

Academic Editor

PLOS ONE

Journal Requirements:

6. PLOS requires an ORCID iD for the corresponding author in Editorial Manager on papers submitted after December 6th, 2016. Please ensure that you have an ORCID iD and that it is validated in Editorial Manager. To do this, go to ‘Update my Information’ (in the upper left-hand corner of the main menu), and click on the Fetch/Validate link next to the ORCID field. This will take you to the ORCID site and allow you to create a new iD or authenticate a pre-existing iD in Editorial Manager.

Reviewers' comments:

Reviewer's Responses to Questions

**Comments to the Author**

1. Is the manuscript technically sound, and do the data support the conclusions?

Reviewer #1: Partly

Reviewer #2: Yes

Reviewer #3: Partly

2. Has the statistical analysis been performed appropriately and rigorously?

Reviewer #1: Yes

Reviewer #2: Yes

Reviewer #3: Yes

3. Have the authors made all data underlying the findings in their manuscript fully available?

Reviewer #1: Yes

Reviewer #2: Yes

Reviewer #3: Yes

4. Is the manuscript presented in an intelligible fashion and written in standard English?

Reviewer #1: Yes

Reviewer #2: Yes

Reviewer #3: Yes

5. Review Comments to the Author

Reviewer #1: The manuscript presents a valuable contribution to the field of bearing fault detection. However, it requires significant revisions to enhance its clarity, detail, and impact. Addressing the issues outlined in the specific review comments would strengthen the manuscript and its suitability for publication in PLOS ONE. The authors should pay particular attention to the clarity of the mathematical descriptions, the detail and presentation of the experimental section and results, and the discussion of the innovation and practical implications of their method. Upon satisfactory revision, the manuscript could be considered for publication in PLOS ONE.

Specific comments on the revisions can be found in the attachment.

Reviewer #2: This manuscript proposes a new method for early bearing defect identification, which is more accurate in judging defects. Please consider making the following improvements:

1. A large number of formulas are listed in the method section. Are these formulas involved in the processing? The author did not explain how the signal was processed, or even what type of signal it was. Is it a magnetic signal? It should be supplemented and explained in detail how the input data was processed using the improved algorithm.

2. Is the sample size for verifying the effectiveness of the method too small? Is the type of bearing too single?

3. Does the lubrication state of the bearing affect the data processing results? Please explain.

Reviewer #3: This is a classic while still challenging topic. You have done and presented some good work. I will recommend its publishing after satisfactory compulsory updating/revising:

1. Introduction: weak in the statement of the problem, particularly how this method proposed and used in this paper is further developed form existing literature and how it is clearly different from them?

In the introduction section, add a short summary what this has done before claiming its contribution;

2. Method section: reference should be added

3. Discussion: while good results were presented, the discussion is weak and lacks the depth. Why this method worked better?

6. PLOS authors have the option to publish the peer review history of their article (what does this mean?). If published, this will include your full peer review and any attached files.

Reviewer #1: No

Reviewer #2: No

Reviewer #3: **Yes: **Qiang Xu

---

## [Author Response · Author response to Decision Letter 1]

17 Aug 2025

We would like to thank the editor who valued our work and found it appropriate to send this paper for review. Thanks for this opportunity, we had the chance to increase the quality of this paper through the remarks of kind reviewers. All modifications done according to the requests of editor/reviewers are shown in red color in the revised manuscript

Reply to Reviewer #1

Before embarking on the response phase to practical and highly informed remarks, the authors wish to express their gratitude for the time devoted by each reviewer for the careful and informed reading of this paper whose concerns we had found interesting and worthy of the seasoned experts strongly master of the field.

There are some comments or advice as following:

Comment 1: The authors should pay particular attention to the clarity of the mathematical descriptions, the detail and presentation of the experimental section and results, and the discussion of the innovation and practical implications of their method.

Response 1: We are grateful for this constructive comment. In the revised version of the manuscript, we have taken the comments into account as follows:

• Clarity of mathematical descriptions text is provided in red:

author action: The mathematical formulations have been reviewed and reworded to ensure greater readability. Textual explanations now accompany each important equation to facilitate understanding, particularly in the subsections on principal : component analysis space design, FFT space design, and kernel principal component analysis application

• Presentation of the experimental section:

author action: The experimental section has been expanded to provide a more comprehensive description of the test protocol, acquisition parameters, defect configurations, and operating conditions. Furthermore, the associated figure has undergone a meticulous reorganization and captioning, with the objective of engendering a more fluid and informative reading experience.

• Presentation of the experimental section:

author action: A dedicated discussion has been appended to the "Conclusion" section in order to emphasise the innovative contributions of the proposed method. In particular, we highlight the contribution of FFT coupling with PCA/KPCA, which improves diagnostic sensitivity in the principal domain (PCA-FFT) and in the residual domain (KPCA-FFT).

Furthermore, the potential for the application of this methodology to other types of bearings and defects has been outlined, thereby emphasising the robustness and generalisability of the proposed framework within a genuine industrial context.

Reply to Reviewer #2

Comment 1: A large number of formulas are listed in the method section. Are these formulas involved in the processing? The author did not explain how the signal was processed, or even what type of signal it was. Is it a magnetic signal? It should be supplemented and explained in detail how the input data was processed using the improved algorithm.

Response 1: I would like to express my gratitude for this constructive comment.

• Nature of the signal used:

author action: The type of signal used in this study is a vibration signal (accelerometric) from sensors mounted on a test bench dedicated to monitoring ball bearings with various defects. This information has been explicitly added to the experimental description in result and discussion section, specifying the type of sensor, its sampling frequency, and the operating conditions of the bearing .

• Link between equations and actual processing:

author action: All formulas presented in the methodology section are actually implemented in signal processing. They have been grouped into two main stages:

- The extraction of characteristics is achieved through the utilisation of PCA/KPCA decomposition, covariance matrix calculation , encompassing projection formulas

and Gaussian kernel, amongst other methodologies.

- A statistical evaluation of the projections was conducted, which involved the calculation of Hotelling's T² and the SPE error index.

• Description of the signal processing applied:

author action: A specific section has been included to detail the processing procedure, from the initial acquisition of the vibration signal to the determination of the final diagnosis:

- Pre-processing: centering, normalization

- Projection: application of PCA or KPCA to the characteristic vectors.

- Frequency transformation: application of FFT to the principal components to enrich the discriminating information.

- Statistical analysis: a methodical and rigorous approach is necessary to ensure the reliability of the results. This approach involves calculating the T² and SPE indicators and comparing them to established thresholds. This comparison allows for the detection of any anomalies

Comment 2: Is the sample size for verifying the effectiveness of the method too small? Is the type of bearing too single?

Response 2:

author action: In relation to the study's sample size, it is important to acknowledge its inherent limitation within the context of this particular investigation. The primary objective of the study was to validate the proposed methodology in a controlled experimental context, employing simulated data and meticulously characterised defects. However, particular attention was paid to increasing the number of defect cases (variations in width and rotation speed) in order to illustrate the robustness of the approach in several realistic scenarios.

With regard to the type of bearing, the study concentrated on a single-row rigid ball bearing due to its extensive utilisation in industrial applications and its uncomplicated structure, which facilitates effective isolation and the modelling of vibration phenomena. Nevertheless, the methodology developed is based on general principles of frequency analysis, linear and nonlinear projection (PCA/KPCA), and statistical evaluation (T², SPE). These principles can be applied to other types of bearings (e.g. roller bearings, angular contact bearings). It is recommended that future research efforts concentrate on the validation of this method using data from a variety of industrial bearings. This would include real signals obtained from test benches or under actual operating conditions.

Comment 3: Does the lubrication state of the bearing affect the data processing results? Please explain.

Response 3:

author action: It should be noted that the lubrication condition of the bearing has a significant influence on the results of data processing, particularly with regard to the amplitude and nature of the vibration signals analyzed.

Poor lubrication, whether insufficient, excessive, or contaminated, is likely to cause mechanical malfunctions, including:

- increased friction forces.

- a rise in local temperature,

- premature deterioration or microscopic alterations to the surface of the raceways or rolling elements.

- changes in the frequency content of the signal, introducing noise or parasitic components.

Han Peng et al.[1] have shown that lubrication failures, whether due to excess, insufficiency, or contamination, are the cause of nearly 70% of bearing defects, mainly wear and corrosion. These phenomena are influenced by various factors, including physical parameters such as speed, temperature, and load; chemical parameters such as oxidation and the presence of wear particles; and the presence of contaminants.

The work of Fanzhao et al.[2] provides further insight by confirming this influence. Indeed, the numerical analyses and experimental tests conducted by the authors reveal that lubricant viscosity has a significant influence on the vibration response of defective bearings. In more detail, it was established that elevated viscosity results in a decrease in vibration amplitude at the characteristic frequencies of failure, attributable to enhanced energy dissipation within the lubricant film.

The study by Juha Miettinen et al.[3], posits that the presence of solid particles in contaminated lubricating grease exerts a significant influence on the amplitude of the statistical indicators extracted. The authors demonstrated that the values of parameters such as pulse count increase significantly in the presence of solid contaminants, compared to those observed with healthy grease. This observation indicates that the quality of the lubricant has a direct impact on the response of bearing operation, introducing additional irregularities that accentuate the pulses in the signal.

In the present study, it was found that, although the lubrication status was not explicitly taken into account in the model, its effects were implicitly integrated into the collected signals and were subsequently incorporated into the processing process. The methodology employed in this study is based on a combination of principal component analysis and kernel principal component analysis (PCA/KPCA), fast Fourier transform (FFT), and statistical index tests (SPE and Hotelling's T²). This methodology facilitates the capture and analysis of vibration disturbances, whether attributable to geometric defects or variations in operating conditions, such as lubrication.

References

[1] B. Peng, Y. Bi, B. Xue, M. Zhang, et S. Wan, « A Survey on Fault Diagnosis of Rolling Bearings », Algorithms, vol. 15, no 10, p. 347, sept. 2022, doi: 10.3390/a15100347.

[2] F. Kong, W. Huang, Y. Jiang, W. Wang, et X. Zhao, « Research on effect of damping variation on vibration response of defective bearings », Adv. Mech. Eng., vol. 11, no 3, p. 1687814019827733, mars 2019, doi: 10.1177/1687814019827733.

[3] J. Miettinen et P. Andersson, « Acoustic emission of rolling bearings lubricated with contaminated grease », Tribol. Int., vol. 33, no 11, p. 777‑787, nov. 2000, doi: 10.1016/S0301-679X(00)00124-9.

Reply to Reviewer #3

Comment 1: Introduction: weak in the statement of the problem, particularly how this method proposed and used in this paper is further developed form existing literature and how it is clearly different from them?

In the introduction section, add a short summary what this has done before claiming its contribution;

Response 1: We would like to express our gratitude for this pertinent observation concerning the formulation of the problem and the development of the methodological approach in the initial manuscript.

author action: In response to these observations, the problem has been reformulated in order to clarify its originality and to better specify the practical use of the proposed method. This approach has highlighted the innovative aspects of the method in the context of bearing diagnosis in an industrial setting. The improvements made can be summarised as follows:

• Problem statement (Introduction)

As previously highlighted, the diagnosis of bearings in industrial environments remains a complex process, primarily due to the presence of high ambient noise, variability in vibration signals, and the low amplitude of initial faults. The efficacy of conventional methodologies for early fault detection is diminished by these limiting factors.

• Positioning of the method

The proposed approach is based on structural filtering using Principal Component Analysis (PCA) and its nonlinear extension (KPCA), which allows dominant information to be separated from anomalies masked in the signals. This enables a more detailed interpretation of the vibration dynamics in two orthogonal subspaces: principal and residual.

• Methodology (Method section)

The methodological process has been described in detail:

-Joint application of PCA and ACPK to raw signals;

- Separation of subspaces (high and low variance);

- Frequency analysis (FFT) applied separately in each subspace;

- Calculation of statistical indices (Hotelling T2 in the principal space and SPE in the residual space);

- Cross-diagnosis based on the two readings.

• Justification for the methodological choice

The hybrid and innovative nature of the method has been emphasised, and it has been demonstrated that this enables the following:

- Increase detection sensitivity to weak or uncertain defects;

- Process complex or noisy signals from industrial environments;

- Provide an enhanced reading of bearing health.

Comment 2: Method section: reference should be added

Response 2:

author action: A comprehensive revision of the "Method" section has been undertaken, with the objective of incorporating pertinent bibliographic references that substantiate the mathematical formulations employed, such as PCA analysis, KPCA T2 and SPE indicators, in addition to FFT expressions.

Comment 3: Discussion: while good results were presented, the discussion is weak and lacks the depth. Why this method worked better?

Response 3:

author action: The sections entitled "Synthesis" and "Conclusion" have been reinforced to provide detailed justifications for the enhanced efficiency of the proposed method (PCA-FFT and KPCA-FFT) in comparison to conventional approaches. Moreover, the integration of dimension reduction approaches (PCA/KPCA) with the Fast Fourier Transform (FFT) has been demonstrated to result in an augmented sensitivity to abnormal deviations. This property provides enhanced reliability in defect detection, including those of low amplitude. This assertion is reinforced by the presence of statistical indicators such as T2 and SPE.

Thank you for your comments and suggestions

---

## [Decision Letter · Decision Letter 1]

21 Oct 2025

A simplified approach to failure analysis of ball bearings combining Principal Component Analysis and Fast Fourier Transform

PONE-D-25-26658R1

Dear Dr. Kanaa,

We’re pleased to inform you that your manuscript has been judged scientifically suitable for publication and will be formally accepted for publication once it meets all outstanding technical requirements.

Kind regards,

Dandan Peng

Academic Editor

PLOS ONE

Additional Editor Comments (optional):

Reviewers' comments:

Reviewer's Responses to Questions

**Comments to the Author**

1. If the authors have adequately addressed your comments raised in a previous round of review and you feel that this manuscript is now acceptable for publication, you may indicate that here to bypass the “Comments to the Author” section, enter your conflict of interest statement in the “Confidential to Editor” section, and submit your "Accept" recommendation.

Reviewer #1: All comments have been addressed

Reviewer #3: All comments have been addressed

2. Is the manuscript technically sound, and do the data support the conclusions?

Reviewer #1: Partly

Reviewer #3: Yes

3. Has the statistical analysis been performed appropriately and rigorously?

Reviewer #1: Yes

Reviewer #3: N/A

4. Have the authors made all data underlying the findings in their manuscript fully available?

Reviewer #1: Yes

Reviewer #3: No

5. Is the manuscript presented in an intelligible fashion and written in standard English?

Reviewer #1: No

Reviewer #3: Yes

6. Review Comments to the Author

Reviewer #1: (No Response)

Reviewer #3: I am satisfied with the revising on the three aspects which I raised. The revising has improved the quality of the paper for publishing.

7. PLOS authors have the option to publish the peer review history of their article (what does this mean?). If published, this will include your full peer review and any attached files.

Reviewer #1: No

Reviewer #3: **Yes: **Qiang Xu

---

## [Editor Report · Acceptance letter]

PONE-D-25-26658R1

PLOS ONE

Dear Dr. Kanaa,

I'm pleased to inform you that your manuscript has been deemed suitable for publication in PLOS ONE. Congratulations! Your manuscript is now being handed over to our production team.

Kind regards,

on behalf of

Dr. Dandan Peng

Academic Editor

PLOS ONE